# Convergence of Gradient EM on Multi-component Mixture of Gaussians

**Bowei Yan**
University of Texas at Austin
boweiy@utexas.edu

**Mingzhang Yin**
University of Texas at Austin
mzyin@utexas.edu

**Purnamrita Sarkar**
University of Texas at Austin
purna.sarkar@austin.utexas.edu

## Abstract

In this paper, we study convergence properties of the gradient variant of Expectation-Maximization algorithm [11] for Gaussian Mixture Models for arbitrary number of clusters and mixing coefficients. We derive the convergence rate depending on the mixing coefficients, minimum and maximum pairwise distances between the true centers, dimensionality and number of components; and obtain a near-optimal local contraction radius. While there have been some recent notable works that derive local convergence rates for EM in the two symmetric mixture of Gaussians, in the more general case, the derivations need structurally different and non-trivial arguments. We use recent tools from learning theory and empirical processes to achieve our theoretical results.

## 1 Introduction

Proposed by [7] in 1977, the Expectation-Maximization (EM) algorithm is a powerful tool for statistical inference in latent variable models. A famous example is the parameter estimation problem under parametric mixture models. In such models, data is generated from a mixture of a known family of parametric distributions. The mixture component from which a datapoint is generated from can be thought of as a latent variable.

Typically the marginal data log-likelihood (which integrates the latent variables out) is hard to optimize, and hence EM iteratively optimizes a lower bound of it and obtains a sequence of estimators. This consists of two steps. In the expectation step (E-step) one computes the expectation of the complete data likelihood with respect to the posterior distribution of the unobserved mixture memberships evaluated at the current parameter estimates. In the maximization step (M-step) one this expectation is maximized to obtain new estimators. EM always improves the objective function. While it is established in [4] that the true parameter vector is the global maximizer of the log-likelihood function, there has been much effort to understand the behavior of the local optima obtained via EM.

When the exact M-step is burdensome, a popular variant of EM, named Gradient EM is widely used. The idea here is to take a gradient step towards the maxima of the expectation computed in the E-step. [11] introduces a gradient algorithm using one iteration of Newton's method and shows the local properties of the gradient EM are almost identical with those of the EM.

Early literature [22, 24] mostly focuses on the convergence to the stationary points or local optima. In [22] it is proven that the sequence of estimators in EM converges to stationary point when the lower bound function from E-step is continuous. In addition, some conditions are derived under

which EM converges to local maxima instead of saddle points; but these are typically hard to check. A link between EM and gradient methods is forged in [24] via a projection matrix and the local convergence rate of EM is obtained. In particular, it is shown that for GMM with well-separated centers, the EM achieves faster convergence rates comparable to a quasi-Newton algorithm. While the convergence of EM deteriorates under worse separations, it is observed in [20] that the mixture density determined by estimator sequence of EM reflects the sample data well.

In recent years, there has been a renewed wave of interest in studying the behavior of EM especially in GMMs. The global convergence of EM for a mixture of two equal-proportion Gaussian distributions is fully characterized in [23]. For more than two clusters, a negative result on EM and gradient EM being trapped in local minima arbitrarily far away from the global optimum is shown in [9].

For high dimensional GMMs with $M$ components, the parameters are learned via reducing the dimensionality via a random projection in [5]. In [6] the two-round method is proposed, where one first initializes with more than $M$ points, then prune to get one point in every cluster. It is pointed out in this paper that in high dimensional space, when the clusters are well separated, the mixing weight will go to either 0 or 1 after one single update. It is showed in [25, 17] that one can cluster high dimensional sub-gaussian mixtures by semi-definite programming relaxations.

For the convergence rate of EM algorithm, it is observed in [19] that a very small mixing proportion for one mixture component compared to others leads to slow convergence. [2] gives non-asymptotic convergence guarantees in isotropic, balanced, two-component GMM; their result proves the linear convergence of EM if the center is initialized in a small neighborhood of the true parameters. The local convergence result in this paper has a sub-optimal contraction region.

$K$-means clustering is another widely used clustering method. Lloyd's algorithm for $k$-means clustering has a similar flavor as EM. At each step, it recomputes the centroids of each cluster and updates the membership assignments alternatively. While EM does soft clustering at each step, Lloyd's algorithm obtains hard clustering. The clustering error of Lloyd's algorithm for arbitrary number of clusters is studied in [13]. The authors also show local convergence results where the contraction region is less restrictive than [2].

We would like to point out that there are many notable algorithms [10, 1, 21] with provable guarantees for estimating mixture models. In [14, 8] the authors propose polynomial time algorithms which achieve epsilon approximation to the k-means loss. A spectral algorithm for learning mixtures of gaussians is proposed in [21]. We want to point out that our aim is not to come up with a new algorithm for mixture models, but to understand the interplay of model parameters in the convergence of gradient EM for a mixture of Gaussians with $M$ components. As we discuss later, our work also immediately leads to convergence guarantees of Stochastic Gradient EM. Another important difference is that the aim of these works is recovering the hidden mixture component memberships, whereas our goal is completely different: we are interested in understanding how well EM can estimate the mean parameters under a good initialization.

In this paper, we study the convergence rate and local contraction radius of gradient EM under GMM with arbitrary number of clusters and mixing weights which are assumed to be known. For simplicity, we assume that the components share the same covariance matrix, which is known. Thus it suffices to carry out our analysis for isotropic GMMs with identity as the shared covariance matrix. We obtain a near-optimal condition on the contraction region in contrast to [2]'s contraction radius for the mixture of two equal weight Gaussians. We want to point out that, while the authors of [2] provide a general set of conditions to establish local convergence for a broad class of mixture models, the derivation of specific results and conditions on local convergence are tailored to the balance and symmetry of the model.

We follow the same general route: first we obtain conditions for population gradient EM, where all sample averages are replaced by their expected counterpart. Then we translate the population version to the sample one. While the first part is conceptually similar, the general setting calls for more involved analysis. The second step typically makes use of concepts from empirical processes, by pairing up Ledoux-Talagrand contraction type arguments with well established symmetrization results. However, in our case, the function is not a contraction like in the symmetric two component case, since it involves the cluster estimates of all $M$ components. Furthermore, the standard analysis of concentration inequalities by McDiarmid's inequality gets complicated because the bounded difference condition is not satisfied in our setting. We overcome these difficulties by taking advan-

tage of recent tools in Rademacher averaging for vector valued function classes, and variants of McDiarmid type inequalities for functions which have bounded difference with high probability.

The rest of the paper is organized as follows. In Section 2, we state the problem and the notations. In Section 3, we provide the main results in local convergence rate and region for both population and sample-based gradient EM in GMMs. Section 4 and 5 provide the proof sketches of population and sample-based theoretical results, followed by the numerical result in Section 6. We conclude the paper with some discussions.

## 2 Problem Setup and Notations

Consider a GMM with $M$ clusters in $d$ dimensional space, with weights $\boldsymbol{\pi} = (\pi_1, \cdots, \pi_M)$. Let $\boldsymbol{\mu}_i \in \mathbb{R}^d$ be the mean of cluster $i$. Without loss of generality, we assume $\mathbb{E} X = \sum_i \pi_i \boldsymbol{\mu}_i = 0$ and the known covariance matrix for all components is $I_d$. Let $\boldsymbol{\mu} \in \mathbb{R}^{Md}$ be the vector stacking the $\boldsymbol{\mu}_i$s vertically. We represent the mixture as $X \sim \text{GMM}(\boldsymbol{\pi}, \boldsymbol{\mu}, I_d)$, which has the density function $p(x|\boldsymbol{\mu}) = \sum_{i=1}^M \pi_i \phi(x|\boldsymbol{\mu}_i, I_d)$. where $\phi(x; \boldsymbol{\mu}, \Sigma)$ is the PDF of $N(\boldsymbol{\mu}, \Sigma)$. Then the population log-likelihood function as $\mathcal{L}(\boldsymbol{\mu}) = \mathbb{E}_X \log \left( \sum_{i=1}^M \pi_i \phi(X|\boldsymbol{\mu}_i, I_d) \right)$. The Maximum Likelihood Estimator is then defined as $\hat{\boldsymbol{\mu}}_{\text{ML}} = \arg \max p(X|\boldsymbol{\mu})$. EM algorithm is based on using an auxiliary function to lower bound the log likelihood. Define $Q(\boldsymbol{\mu}|\boldsymbol{\mu}^t) = \mathbb{E}_X \left[ \sum_i p(Z = i|X; \boldsymbol{\mu}^t) \log \phi(X; \boldsymbol{\mu}_i, I_d) \right]$, where $Z$ denote the unobserved component membership of data point $X$. The standard EM update is $\boldsymbol{\mu}^{t+1} = \arg \max_{\boldsymbol{\mu}} Q(\boldsymbol{\mu}|\boldsymbol{\mu}^t)$. Define

$$w_i(X; \boldsymbol{\mu}) = \frac{\pi_i \phi(X|\boldsymbol{\mu}_i, I_d)}{\sum_{j=1}^M \pi_j \phi(X|\boldsymbol{\mu}_j, I_d)} \tag{1}$$

The update step for gradient EM, defined via the gradient operator $G(\boldsymbol{\mu}^t) : \mathbb{R}^{Md} \to \mathbb{R}^{Md}$, is

$$G(\boldsymbol{\mu}^t)^{(i)} := \boldsymbol{\mu}_i^{t+1} = \boldsymbol{\mu}_i^t + s[\nabla Q(\boldsymbol{\mu}^t|\boldsymbol{\mu}^t)]_i = \boldsymbol{\mu}_i^t + s\mathbb{E}_X \left[ \pi_i w_i(X; \boldsymbol{\mu}^t)(X - \boldsymbol{\mu}_i^t) \right]. \tag{2}$$

where $s > 0$ is the step size and $(.)^{(i)}$ denotes the part of the stacked vector corresponding to the $i^{th}$ mixture component. We will also use $G_n(\boldsymbol{\mu})$ to denote the empirical counterpart of the population gradient operator $G(\mu)$ defined in Eq (2). We assume we are given an initialization $\boldsymbol{\mu}_i^0$ and the true mixing weight $\pi_i$ for each component.

### 2.1 Notations

Define $R_{\max}$ and $R_{\min}$ as the largest and smallest distance between cluster centers i.e., $R_{\max} = \max_{i \neq j} \|\boldsymbol{\mu}_i^* - \boldsymbol{\mu}_j^*\|$, $R_{\min} = \min_{i \neq j} \|\boldsymbol{\mu}_i^* - \boldsymbol{\mu}_j^*\|$. Let $\pi_{\max}$ and $\pi_{\min}$ be the maximal and minimal cluster weights, and define $\kappa$ as $\kappa = \frac{\pi_{\max}}{\pi_{\min}}$. Standard complexity analysis notation $o(\cdot), O(\cdot), \Theta(\cdot), \Omega(\cdot)$ will be used. $f(n) = \tilde{\Omega}(g(n))$ is short for $\Omega(g(n))$ ignoring logarithmic factors, equivalent to $f(n) \geq Cg(n) \log^k(g(n))$, similar for others. We use $\otimes$ to represent the kronecker product.

## 3 Main Results

Despite being a non-convex problem, EM and gradient EM algorithms have been shown to exhibit good convergence behavior in practice, especially with good initializations. However, existing local convergence theory only applies for two-cluster equal-weight GMM. In this section, we present our main result in two parts. First we show the convergence rate and present a near-optimal radius for contraction region for population gradient EM. Then in the second part we connect the population version to finite sample results using concepts from empirical processes and learning theory.

### 3.1 Local contraction for population gradient EM

Intuitively, when $\boldsymbol{\mu}^t$ equals the ground truth $\boldsymbol{\mu}^*$, then the $Q(\boldsymbol{\mu}|\boldsymbol{\mu}^*)$ function will be well-behaved. This function is a key ingredient in [2], where the curvature of the $Q(\cdot|\boldsymbol{\mu})$ function is shown to be close to the curvature of $Q(\cdot|\boldsymbol{\mu}^*)$ when the $\boldsymbol{\mu}$ is close to $\boldsymbol{\mu}^*$. This is a local property that only requires the gradient to be stable at one point.

**Definition 1** (Gradient Stability). *The Gradient Stability (GS) condition, denoted by $GS(\gamma, a)$, is satisfied if there exists $\gamma > 0$, such that for $\boldsymbol{\mu}_i^t \in \mathbb{B}(\boldsymbol{\mu}_i^*, a)$ with some $a > 0$, for $\forall i \in [M]$.*

$$\|\nabla Q(\boldsymbol{\mu}^t|\boldsymbol{\mu}^*) - \nabla Q(\boldsymbol{\mu}^t|\boldsymbol{\mu}^t)\| \leq \gamma \|\boldsymbol{\mu}^t - \boldsymbol{\mu}^*\|$$

The GS condition is used to prove contraction of the sequence of estimators produced by population gradient EM. However, for most latent variable models, it is typically challenging to verify the GS condition and obtain a tight bound on the parameter $\gamma$. We derive the GS condition under milder conditions (see Theorem 4 in Section 4), which bounds the deviation of the partial gradient evaluated at $\boldsymbol{\mu}_i^t$ uniformly over all $i \in [M]$. This immediately implies the global GS condition defined in Definition 1. Equipped with this result, we achieve a nearly optimal local convergence radius for general GMMs in Theorem 1. The proof of this theorem can be found in Appendix B.2.

**Theorem 1** (Convergence for Population gradient EM). *Let $d_0 := \min\{d, M\}$. If $R_{\min} = \tilde{\Omega}(\sqrt{d_0})$, with initialization $\boldsymbol{\mu}^0$ satisfying, $\|\boldsymbol{\mu}_i^0 - \boldsymbol{\mu}_i^*\| \leq a, \forall i \in [M]$, where*

$$a \leq \frac{R_{\min}}{2} - \sqrt{d_0} O\left(\sqrt{\log\left(\max\left\{\frac{M^2\kappa}{\pi_{\min}}, R_{\max}, d_0\right\}\right)}\right)$$

*then the Population EM converges:*

$$\|\boldsymbol{\mu}^t - \boldsymbol{\mu}^*\| \leq \zeta^t \|\boldsymbol{\mu}_0 - \boldsymbol{\mu}^*\|, \qquad \zeta = \frac{\pi_{\max} - \pi_{\min} + 2\gamma}{\pi_{\max} + \pi_{\min}} < 1$$

*where $\gamma = M^2(2\kappa + 4)(2R_{\max} + d_0)^2 \exp\left(-\left(\frac{R_{\min}}{2} - a\right)^2 \sqrt{d_0}/8\right) < \pi_{\min}$.*

**Remark 1.** *The local contraction radius is largely improved compared to that in [2], which has $R_{\min}/8$ in the two equal sized symmetric GMM setting. It can be seen that in Theorem 1, $a/R_{\min}$ goes to $\frac{1}{2}$ as the signal to noise ratio goes to infinity. We will show in simulations that when initialized from some point that lies $R_{\min}/2$ away from the true center, gradient EM only converges to a stationary point which is not a global optimum. More discussion can be found in Section 6.*

### 3.2 Finite sample bound for gradient EM

In the finite sample setting, as long as the deviation of the sample gradient from the population gradient is uniformly bounded, the convergence in the population setting implies the convergence in finite sample scenario. Thus the key ingredient in the proof is to get this uniform bound over all parameters in the contraction region $\mathbb{A}$, i.e. bound $\sup_{\boldsymbol{\mu}\in\mathbb{A}} \|G^{(i)}(\boldsymbol{\mu}) - G_n^{(i)}(\boldsymbol{\mu})\|$, where $G$ and $G_n$ are defined in Section 2.

To prove the result, we expand the difference and define the following function for $i \in [M]$, where $u$ is a unit vector on a $d$ dimensional sphere $\mathcal{S}^{d-1}$. This appears because we can write the Euclidean norm of any vector $B$, as $\|B\| = \sup_{u\in\mathcal{S}^{d-1}} \langle B, u \rangle$.

$$g_i^u(X) = \sup_{\boldsymbol{\mu}\in\mathbb{A}} \frac{1}{n} \sum_{i=1}^n w_1(X_i; \boldsymbol{\mu})\langle X_i - \boldsymbol{\mu}_1, u \rangle - \mathbb{E}w_1(X; \boldsymbol{\mu})\langle X - \boldsymbol{\mu}_1, u \rangle. \qquad (3)$$

We will drop the super and subscript and prove results for $g_1^u$ without loss of generality.

The outline of the proof is to show that $g(X)$ is close to its expectation. This expectation can be further bounded via the Rademacher complexity of the corresponding function class (defined below in Eq (4)) by the tools like the symmetrization lemma [18].

Consider the following class of functions indexed by $\boldsymbol{\mu}$ and some unit vector on $d$ dimensional sphere $u \in \mathcal{S}^{d-1}$:

$$\mathcal{F}_i^u = \{f^i : \mathcal{X} \to \mathbb{R} | f^i(X; \boldsymbol{\mu}, u) = w_i(X; \boldsymbol{\mu})\langle X - \boldsymbol{\mu}_i, u \rangle\} \qquad (4)$$

We need to bound the $M$ functions classes separately for each mixture. Given a finite $n$-sample $(X_1, \cdots, X_n)$, for each class, we define the Rademacher complexity as the expectation of empirical

Rademacher complexity.

$$\hat{R}_n(\mathcal{F}_i^u) = \mathbb{E}_\epsilon \left[ \sup_{\boldsymbol{\mu} \in \mathbb{A}} \frac{1}{n} \sum_{j=1}^n \epsilon_i f^i(X_j; \boldsymbol{\mu}, u) \right]; \qquad R_n(\mathcal{F}_i^u) = \mathbb{E}_X \hat{R}_n(\mathcal{F}_i^u)$$

where $\epsilon_i$'s are the i.i.d. Rademacher random variables.

For many function classes, the computation of the empirical Rademacher complexity can be hard. For complicated functions which are Lipschitz w.r.t functions from a simpler function class, one can use Ledoux-Talagrand type contraction results [12]. In order to use the Ledoux-Talagrand contraction, one needs a 1-Lipschitz function, which we do not have, because our function involves $\boldsymbol{\mu}_i$, $i \in [M]$. Also, the weight functions $w_i$ are not separable in terms of the $\boldsymbol{\mu}_i$'s. Therefore the classical contraction lemma does not apply. In our analysis, we need to introduce a vector-valued function, with each element involving only one $\boldsymbol{\mu}_i$, and apply a recent result of vector-versioned contraction lemma [15]. With some careful analysis, we get the following. The details are deferred to Section 5.

**Proposition 1.** *Let $\mathcal{F}_i^u$ as defined in Eq. (4) for $\forall i \in [M]$, then for some universal constant c,*

$$R_n(\mathcal{F}_i^u) \leq \frac{cM^{3/2}(1 + R_{\max})^3 \sqrt{d} \max\{1, \log(\kappa)\}}{\sqrt{n}}$$

After getting the Rademacher complexity, one needs to use concentration results like McDiarmid's inequality [16] to achieve the finite-sample bound. Unfortunately for the functions defined in Eq. (4), the martingale difference sequence does not have bounded differences. Hence it is difficult to apply McDiarmid's inequality in its classical form. To resolve this, we instead use an extension of McDiarmid's inequality which can accommodate sequences which have bounded differences with high probability [3].

**Theorem 2** (Convergence for sample-based gradient EM). *Let $\zeta$ be the contraction parameter in Theorem 1, and*

$$\epsilon^{unif}(n) = \tilde{O}(\max\{n^{-1/2}M^3(1 + R_{\max})^3 \sqrt{d} \max\{1, \log(\kappa)\}, (1 + R_{\max})d/\sqrt{n}\}). \qquad (5)$$

*If $\epsilon^{unif}(n) \leq (1 - \zeta)a$, then sample-based gradient EM satisfies*

$$\left\| \hat{\boldsymbol{\mu}}_i^t - \boldsymbol{\mu}_i^* \right\| \leq \zeta^t \left\| \boldsymbol{\mu}^0 - \boldsymbol{\mu}^* \right\|_2 + \frac{1}{1 - \zeta} \epsilon^{unif}(n); \quad \forall i \in [M]$$

*with probability at least $1 - n^{-cd}$, where c is a positive constant.*

**Remark 2.** *When data is observed in a streaming fashion, the gradient update can be modified into a stochastic gradient update, where the gradient is evaluated based on a single observation or a small batch. By the GS condition proved in Theorem 1, combined with Theorem 6 in [2], we immediately extend the guarantees of gradient EM into the guarantees for the stochastic gradient EM.*

### 3.3 Initialization

Appropriate initialization for EM is the key to getting good estimation within fewer restarts in practice. There have been a number of interesting initialization algorithms for estimating mixture models. It is pointed out in [9] that in practice, initializing the centers by uniformly drawing from the data is often more reasonable than drawing from a fixed distribution. Under this initialization strategy, we can bound the number of initializations required to find a "good" initialization that falls in the contraction region in Theorem 1. The exact theorem statement and a discussion of random initialization can be found in Appendix D. More sophisticated strategy includes, an approximate solution to $k$-means on a projected low-dimensional space used in [1] and [10]. While it would be interesting to study different initialization schemes, that is part of future work.

## 4 Local Convergence of Population Gradient EM

In this section we present the proof sketch for Theorem 1. The complete proofs in this section are deferred to Appendix B. To start with, we calculate the closed-form characterization of the gradient of $q(\boldsymbol{\mu})$ as stated in the following lemma.

**Lemma 1.** *Define $q(\boldsymbol{\mu}) = Q(\boldsymbol{\mu}|\boldsymbol{\mu}^*)$. The gradient of $q(\boldsymbol{\mu})$ is $\nabla q(\boldsymbol{\mu}) = (diag(\pi) \otimes I_d)(\boldsymbol{\mu}^* - \boldsymbol{\mu})$.*

If we know the parameter $\gamma$ in the gradient stability condition, then the convergence rate depends only on the condition number of the Hessian of $q(\cdot)$ and $\gamma$.

**Theorem 3** (Convergence rate for population gradient EM). *If $Q$ satisfies the GS condition with parameter $0 < \gamma < \pi_{\min}$, denote $d_t := \|\boldsymbol{\mu}_t - \boldsymbol{\mu}^*\|$, then with step size $s = \frac{2}{\pi_{\min} + \pi_{\max}}$, we have:*

$$d_{t+1} \le \left( \frac{\pi_{\max} - \pi_{\min} + 2\gamma}{\pi_{\max} + \pi_{\min}} \right)^t d_0$$

The proof uses an approximation on gradient and standard techniques in analysis of gradient descent.

**Remark 3.** *It can be verified that the convergence rate is equivalent to that shown in [2] when applied to GMMs. The convergence slows down as the proportion imbalance $\kappa = \pi_{\max}/\pi_{\min}$ increases, which matches the observation in [19].*

Now to verify the GS condition, we have the following theorem.

**Theorem 4** (GS condition for general GMM). *If $R_{\min} = \tilde{\Omega}(\sqrt{\min\{d, M\}})$, and $\boldsymbol{\mu}_i \in \mathbb{B}(\boldsymbol{\mu}_i^*, a), \forall i \in [M]$ where $a \le \frac{R_{\min}}{2} - \sqrt{\min\{d, M\}} \max(4\sqrt{2[\log(R_{\min}/4)]_+}, 8\sqrt{3})$, then $\|\nabla_{\boldsymbol{\mu}_i} Q(\boldsymbol{\mu}|\boldsymbol{\mu}^t) - \nabla_{\boldsymbol{\mu}_i} q(\boldsymbol{\mu})\| \le \frac{\gamma}{M} \sum_{i=1}^{M} \|\boldsymbol{\mu}_i^t - \boldsymbol{\mu}_i^*\| \le \frac{\gamma}{\sqrt{M}} \|\boldsymbol{\mu}^t - \boldsymbol{\mu}^*\|$,*
*where $\gamma = M^2(2\kappa + 4)(2R_{\max} + \min\{d, M\})^2 \exp\left( -\left(\frac{R_{\min}}{2} - a\right)^2 \sqrt{\min\{d, M\}}/8 \right)$.*
*Furthermore, $\|\nabla Q(\boldsymbol{\mu}|\boldsymbol{\mu}^t) - \nabla q(\boldsymbol{\mu})\| \le \gamma \|\boldsymbol{\mu}^t - \boldsymbol{\mu}^*\|$.*

*Proof sketch of Theorem 4.* W.l.o.g. we show the proof with the first cluster, consider the difference of the gradient corresponding to $\boldsymbol{\mu}_1$.

$$\nabla_{\boldsymbol{\mu}_1} Q(\boldsymbol{\mu}^t|\boldsymbol{\mu}^t) - \nabla_{\boldsymbol{\mu}_1} q(\boldsymbol{\mu}^t) = \mathbb{E}(w_1(X; \boldsymbol{\mu}^t) - w_1(X; \boldsymbol{\mu}^*))(X - \boldsymbol{\mu}_1^t) \tag{6}$$

For any given $X$, consider the function $\boldsymbol{\mu} \to w_1(X; \boldsymbol{\mu})$, we have

$$\nabla_{\boldsymbol{\mu}} w_1(X; \boldsymbol{\mu}) = \begin{pmatrix} w_1(X; \boldsymbol{\mu})(1 - w_1(X; \boldsymbol{\mu}))(X - \boldsymbol{\mu}_1)^T \\ -w_1(X; \boldsymbol{\mu}) w_2(X; \boldsymbol{\mu})(X - \boldsymbol{\mu}_2)^T \\ \vdots \\ -w_1(X; \boldsymbol{\mu}) w_M(X; \boldsymbol{\mu})(X - \boldsymbol{\mu}_M)^T \end{pmatrix} \tag{7}$$

Let $\boldsymbol{\mu}^u = \boldsymbol{\mu}^* + u(\boldsymbol{\mu}^t - \boldsymbol{\mu}^*), \forall u \in [0, 1]$, obviously $\boldsymbol{\mu}^u \in \Pi_{i=1}^M \mathbb{B}(\boldsymbol{\mu}_i^*, \|\boldsymbol{\mu}_i^t - \boldsymbol{\mu}_i^*\|) \subset \Pi_{i=1}^M \mathbb{B}(\boldsymbol{\mu}_i^*, a)$. By Taylor's theorem,

$$\|\mathbb{E}(w_1(X; \boldsymbol{\mu}_1^t) - w_1(X; \boldsymbol{\mu}_1^*))(X - \boldsymbol{\mu}_1^t)\| = \left\| \mathbb{E}\left[ \int_{u=0}^{1} \nabla_u w_1(X; \boldsymbol{\mu}^u) du (X - \boldsymbol{\mu}_1^t) \right] \right\|$$
$$\le U_1 \|\boldsymbol{\mu}_1^t - \boldsymbol{\mu}_1^*\|_2 + \sum_{i \ne 1} U_i \|\boldsymbol{\mu}_i^t - \boldsymbol{\mu}_i^*\|_2 \le \max_{i \in [M]}\{U_i\} \sum_i \|\boldsymbol{\mu}_i^t - \boldsymbol{\mu}_i^*\|_2 \tag{8}$$

where

$$U_1 = \sup_{u \in [0,1]} \|\mathbb{E} w_1(X; \boldsymbol{\mu}^u)(1 - w_1(X; \boldsymbol{\mu}^u))(X - \boldsymbol{\mu}_1^t)(X - \boldsymbol{\mu}_1^u)^T\|_{op}$$

$$U_i = \sup_{u \in [0,1]} \|\mathbb{E} w_1(X; \boldsymbol{\mu}^u) w_i(X; \boldsymbol{\mu}^u)(X - \boldsymbol{\mu}_1^t)(X - \boldsymbol{\mu}_2^u)^T\|_{op}$$

Bounding them with careful analysis on Gaussian distribution yields the result. The technical details are deferred to Appendix B. $\qquad\square$

## 5  Sample-based Convergence

In this section we present the proof sketch for sample-based convergence of gradient EM. The main ingredient in proof of Theorem 2 is the result of the following theorem, which develops an uniform upper bound for the differences between sample-based gradient and population gradient on each cluster center.

**Theorem 5** (Uniform bound for sample-based gradient EM). *Denote $\mathbb{A}$ as the contraction region $\Pi_{i=1}^{M}\mathbb{B}(\boldsymbol{\mu}_i^*, a)$. Under the condition of Theorem 1, with probability at least $1 - \exp(-cd\log n)$,*

$$\sup_{\boldsymbol{\mu}\in\mathbb{A}} \left\| G^{(i)}(\boldsymbol{\mu}) - G_n^{(i)}(\boldsymbol{\mu}) \right\| < \epsilon^{unif}(n); \qquad \forall i \in [M]$$

*where $\epsilon^{unif}(n)$ is as defined in Eq. (5).*

**Remark 4.** *It is worth pointing out that, the first part of the bound on $\epsilon^{unif}(n)$ in Eq. (5) comes from the Rademacher complexity, which is optimal in terms of the order of $n$ and $d$. However the extra factor of $\sqrt{d}$ and $\log(n)$ comes from the altered McDiarmid's inequality, tightening which will be left for future work.*

*Proof sketch of Theorem 5.* Denote $Z_i = \sup_{\boldsymbol{\mu}\in\mathbb{A}} \left\| G^{(i)}(\boldsymbol{\mu}) - G_n^{(i)}(\boldsymbol{\mu}) \right\|$. Recall $g_i^u(X)$ defined in Eq. (3), then it is not hard to see that $Z_i = \sup_{u\in\mathcal{S}^{d-1}} g_i^u(X)$. Consider a $\frac{1}{2}$-covering $\{u^{(1)}, \cdots, u^{(K)}\}$ of the unit sphere $\mathcal{S}^{d-1}$, where $K$ is the covering number of an unit sphere in $d$ dimensions. We can show that $Z_i \le 2 \max_{j=1,\cdots,K} g_i^{u^{(j)}}(X)$.

As we state below in Lemma 2, we have for each $u$, with probability at least $1 - \exp(-cd\log n)$, $g_i^u(X) = \tilde{O}(\max\{R_n(\mathcal{F}_i^u), (1 + R_{\max})d/\sqrt{n}\})$. Plugging in the Rademacher complexity from Proposition 1, standard bounds on $K$, and applying union bound, we have

$$Z_i \le \tilde{O}(\max\{n^{-1/2}M^3(1 + R_{\max})^3\sqrt{d}\max\{1, \log(\kappa)\}, (1 + R_{\max})d/\sqrt{n}\})$$

with probability at least $1 - \exp(2d - cd\log n) = 1 - \exp(-c'd\log n)$. $\qquad\square$

Iteratively applying Theorem 5, we can bound the error in $\boldsymbol{\mu}$ for the sample updates. The full proof is deferred to Appendix C. The key step is the following lemma, which bounds the $g_i^u(X)$ for any given $u \in \mathcal{S}^{d-1}$. Without loss of generality we prove the result for $i = 1$.

**Lemma 2.** *Let $u$ be a unit vector. $X_i, i = 1, \cdots, n$ are i.i.d. samples from $GMM(\pi, \boldsymbol{\mu}^*, I_d)$. $g_1^u(X)$ as defined in Eq. (3). Then with probability $1 - \exp(-cd\log n)$ for some constant $c > 0$, $g_1^u(X) = \tilde{O}(\max\{R_n(\mathcal{F}_1^u), (1 + R_{\max})d/\sqrt{n}\})$.*

The quantity $g_1^u(X)$ depends on the sample, the idea for proving Lemma 2 is to show it concentrates around its expectation when sample size is large. Note that when the function class has bounded differences (changing one data point changes the function by a bounded amount almost surely), as in the case in many risk minimization problems in supervised learning, the McDiarmid's inequality can be used to achieve concentration. However the function class we define in Eq. (4) is not bounded almost everywhere, but with high probability, hence the classical result does not apply. Conditioning on the event where the difference is bounded, we use an extension of McDiarmid's inequality by [3]. For convenience, we restate a weaker version of the theorem using our notation below.

**Theorem 6** ([3]). *Consider independent random variable $X = (X_1, \cdots, X_n)$ in the product probability space $\mathcal{X} = \prod_{i=1}^{n}\mathcal{X}_i$, where $\mathcal{X}_i$ is the probability space for $X_i$. Also consider a function $g : \mathcal{X} \to \mathbb{R}$. If there exists a subset $\mathcal{Y} \subset \mathcal{X}$, and a scalar $c > 0$, such that*

$$|g(x) - g(y)| \le L, \forall x, y \in \mathcal{Y}, x_j = y_j, \forall j \ne i.$$

*Denote $p = 1 - P(X \in \mathcal{Y})$, then $P(g(X) - \mathbb{E}[g(X)|X \in \mathcal{Y}] \ge \epsilon) \le p + \exp\left(-\frac{2(\epsilon - npL)_+^2}{nL^2}\right)$.*

It is worth pointing out that in Theorem 6, the concentration is shown in reference to the conditional expectation $\mathbb{E}[g(X)|X \in \mathcal{Y}]$ when the data points are in the bounded difference set. So to fully achieve the type of bound given by McDiarmid's inequality, we need to further bound the difference of the conditional expectation and the full expectation. Combining the two parts we will be able to show that, the empirical difference is upper bounded using the Rademacher complexity.

Now it remains to derive the Rademacher complexity under the given function class. Note that when the function class is a contraction, or Lipschitz with respect to another function (usually of a simpler form), one can use the Ledoux-Talagrand contraction lemma [12] to reduce the Rademacher complexity of the original function class to the Rademacher complexity of the simpler function class. This is essential in getting the Rademacher complexities for complicated function classes. As

we mention in Section 3, our function class in Eq. (4) is unfortunately not Lipschitz due to the fact that it involves all cluster centers even for the gradient on one cluster. We get around this problem by introducing a vector valued function, and show that the functions in Eq. (4) are Lipschitz in terms of the vector-valued function. In other words, the absolute difference in the function when the parameter changes is upper bounded by the norm of the vector difference of the vector-valued function. Then we build upon the recent vector-contraction result from [15], and prove the following lemma under our setting.

**Lemma 3.** *Let $X$ be nontrivial, symmetric and sub-gaussian. Then there exists a constant $C < \infty$, depending only on the distribution of $X$, such that for any subset $\mathcal{S}$ of a separable Banach space and function $h_i : \mathcal{S} \to \mathbb{R}$, $f_i : \mathcal{S} \to \mathbb{R}^k$, $i \in [n]$ satisfying $\forall s, s' \in \mathcal{S}$, $|h_i(s) - h_i(s')| \leq L\|f(s) - f(s')\|$. If $\epsilon_{ik}$ is an independent doubly indexed Rademacher sequence, we have,*

$$\mathbb{E} \sup_{s \in \mathcal{S}} \sum_i \epsilon_i h_i(s) \leq \mathbb{E} \sqrt{2} L \sup_{s \in \mathcal{S}} \sum_{i,k} \epsilon_{ik} f_i(s)_k,$$

*where $f_i(s)_k$ is the k-th component of $f_i(s)$.*

**Remark 5.** *In contrast to the original form in [15], we have a $\mathcal{S}$ as a subset of a separable Banach Space. The proof uses standard tools from measure theory, and is to be found in Appendix C.*

This equips us to prove Proposition 1.

*Proof sketch of Proposition 1.* For any unit vector $u$, the Rademacher complexity of $\mathcal{F}_1^u$ is

$$
\begin{aligned}
R_n(\mathcal{F}_1^u) =& \mathbb{E}_X \mathbb{E}_\epsilon \sup_{\boldsymbol{\mu} \in \mathbb{A}} \frac{1}{n} \sum_{i=1}^n \epsilon_i w_1(X_i; \boldsymbol{\mu}) \langle X_i - \boldsymbol{\mu}_1, u \rangle \\
\leq& \underbrace{\mathbb{E}_X \mathbb{E}_\epsilon \sup_{\boldsymbol{\mu} \in \mathbb{A}} \frac{1}{n} \sum_{i=1}^n \epsilon_i w_1(X_i; \boldsymbol{\mu}) \langle X_i, u \rangle}_{(D)} + \underbrace{\mathbb{E}_X \mathbb{E}_\epsilon \sup_{\boldsymbol{\mu} \in \mathbb{A}} \frac{1}{n} \sum_{i=1}^n \epsilon_i w_1(X_i; \boldsymbol{\mu}) \langle \boldsymbol{\mu}_1, u \rangle}_{(E)}
\end{aligned}
\tag{9}
$$

We bound the two terms separately. Define $\eta_j(\boldsymbol{\mu}) : \mathbb{R}^{Md} \to \mathbb{R}^M$ to be a vector valued function with the $k$-th coordinate

$$[\eta_j(\boldsymbol{\mu})]_k = \frac{\|\boldsymbol{\mu}_1\|^2}{2} - \frac{\|\boldsymbol{\mu}_k\|^2}{2} + \langle X_j, \boldsymbol{\mu}_k - \boldsymbol{\mu}_1 \rangle + \log\left(\frac{\pi_k}{\pi_1}\right)$$

It can be shown that
$$|w_1(X_j; \boldsymbol{\mu}) - w_1(X_j; \boldsymbol{\mu}')| \leq \frac{\sqrt{M}}{4} \|\eta_j(\boldsymbol{\mu}) - \eta_j(\boldsymbol{\mu}')\| \tag{10}$$

Now let $\psi_1(X_j; \boldsymbol{\mu}) = w_1(X_j; \boldsymbol{\mu}) \langle X_j, u \rangle$. With Lipschitz property (10) and Lemma C.1, we have

$$\mathbb{E}\left[\sup_{\boldsymbol{\mu} \in \mathbb{A}} \frac{1}{n} \sum_{j=1}^n \epsilon_j w_i(X_j; \boldsymbol{\mu}) \langle X_j, u \rangle\right] \leq \mathbb{E}\left[\frac{\sqrt{2}\sqrt{M}}{4n} \sup_{\boldsymbol{\mu} \in \mathbb{A}} \sum_{j=1}^n \sum_{k=1}^M \epsilon_{jk} [\eta_j(\boldsymbol{\mu})]_k\right]$$

The right hand side can be bounded with tools regarding independent sum of sub-gaussian random variables. Similar techniques apply to the $(E)$ term. Adding things up we get the final bound. □

## 6 Experiments

In this section we collect some numerical results. In all experiments we set the covariance matrix for each mixture component as identity matrix $I_d$ and define signal-to-noise ratio (SNR) as $R_{\min}$.

**Convergence Rate** We first evaluate the convergence rate and compare with those given in Theorem 3 and Theorem 4. For this set of experiments, we use a mixture of 3 Gaussians in 2 dimensions. In both experiments $R_{\max}/R_{\min} = 1.5$. In different settings of $\boldsymbol{\pi}$, we apply gradient EM with varying SNR from 1 to 5. For each choice of SNR, we perform 10 independent trials with $N = 12,000$

data points. The average of $\log \|\boldsymbol{\mu}^t - \hat{\boldsymbol{\mu}}\|$ and the standard deviation are plotted versus iterations. In Figure 1 (a) and (b) we plot balanced $\boldsymbol{\pi}$ ($\kappa = 1$) and unbalanced $\boldsymbol{\pi}$ ($\kappa > 1$) respectively.

All settings indicate the linear convergence rate as shown in Theorem 3. As SNR grows, the parameter $\gamma$ in GS condition decreases and thus yields faster convergence rate. Comparing left two panels in Figure 1, increasing imbalance of cluster weights $\kappa$ slows down the local convergence rate as shown in Theorem 3.

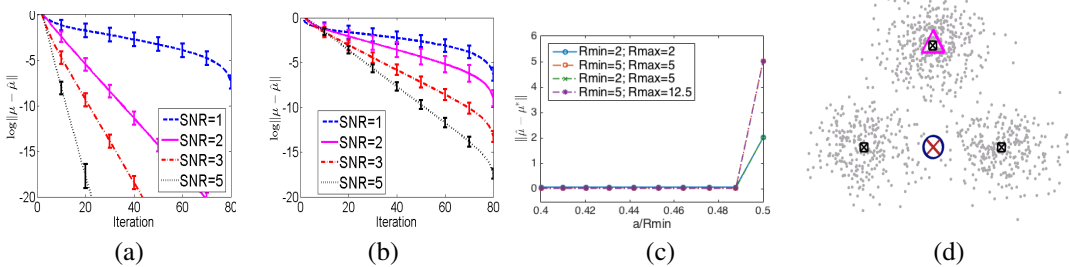

(a)          (b)          (c)          (d)

Figure 1: (a, b): The influence of SNR on optimization error in different settings. The figures represent the influence of SNR when the GMMs have different cluster centers and weights: (a) $\boldsymbol{\pi} = (1/3, 1/3, 1/3)$. (b) $\boldsymbol{\pi} = (0.6, 0.3, 0.1)$. (c) plots statistical error with different initializations arbitrarily close to the boundary of the contraction region. (d) shows the suboptimal stationary point when two centers are initialized from the midpoint of the respective true cluster centers.

**Contraction Region** To show the tightness of the contraction region, we generate a mixture with $M = 3, d = 2$, and initialize the clusters as follows. We use $\boldsymbol{\mu}_2^0 = \frac{\boldsymbol{\mu}_2^* + \boldsymbol{\mu}_3^*}{2} - \epsilon$, $\boldsymbol{\mu}_3^0 = \frac{\boldsymbol{\mu}_2^* + \boldsymbol{\mu}_3^*}{2} + \epsilon$, for shrinking $\epsilon$, i.e. increasing $a/R_{\min}$ and plot the error on the Y axis. Figure 1-(c) shows that gradient EM converges when initialized arbitrarily close to the boundary, thus confirming our near optimal contraction region. Figure 1-(d) shows that when $\epsilon = 0$, i.e. $a = \frac{R_{\min}}{2}$, gradient EM can be trapped at a sub-optimal stationary point.

# 7 Concluding Remarks

In this paper, we obtain local convergence rates and a near optimal contraction radius for population and sample-based gradient EM for multi-component GMMs with arbitrary mixing weights. For simplicity, we assume that the the mixing weights are known, and the covariance matrices are the same across components and known. For our sample-based analysis, we face new challenges where bears structural differences from the two-component, equal-weight setting, which are alleviated via the usage of non-traditional tools like a vector valued contraction argument and martingale concentrations bounds where bounded differences hold only with high probability.

# Acknowledgments

PS was partially supported by NSF grant DMS 1713082.

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
