[Supplementary Material]

# Supplementary Materials for
## *Convergence of Gradient EM on Multi-component Mixture of Gaussians*

**Bowei Yan**
University of Texas at Austin
`boweiy@utexas.edu`

**Mingzhang Yin**
University of Texas at Austin
`mzyin@utexas.edu`

**Purnamrita Sarkar**
University of Texas at Austin
`purna.sarkar@austin.utexas.edu`

This document collects some supplementary materials in *Convergence of Gradient EM on Multi-component Mixture of Gaussians*, including the proofs for the lemmas and theorems presented in that paper.

## A Accompanying Lemmas

In this subsection, we collect some lemmas on Gaussian distribution and basic properties of Gaussian mixture model. Most of them can be derived with fundamental analysis techniques. The following lemma from [7] bounds the covering number of a unit sphere.

**Lemma A.1** (Lemma 5.2 [7])**.** *Let $\mathcal{S}^{n-1}$ be the unit Euclidean sphere equipped with Euclidean metric. Denote $\mathcal{N}(\mathcal{S}^{n-1}, \epsilon)$ as the covering number with $\epsilon$-net, then*

$$\mathcal{N}(\mathcal{S}^{n-1}, \epsilon) \leq \left(1 + \frac{2}{\epsilon}\right)^n$$

*Specifically, when $\epsilon = 1/2$, we have*

$$\mathcal{N}(\mathcal{S}^{n-1}, \frac{1}{2}) \leq \exp(2n)$$

The following lemma is useful while carrying out spherical coordinate transformation.

**Lemma A.2.**      *(1) The volume for a $d$-dimensional $r$-ball is $\frac{\pi^{\frac{d}{2}}}{\Gamma\left(\frac{d}{2}+1\right)} r^d$;*

*(2) $\int_0^\pi \sin^k(x)dx = \frac{\sqrt{\pi}\Gamma\left(\frac{k+1}{2}\right)}{\Gamma\left(\frac{k}{2}+1\right)}$, and*

$$\int_{\theta_{d-1}=0}^{2\pi} \int_{\theta_{d-2}=0}^{\pi} \cdots \int_{\theta_1=0}^{\pi} \sin^{d-2}(\theta_1)\cdots\sin(\theta_{d-2})d\theta_1\cdots d\theta_{d-1} = \frac{2\pi^{\frac{d}{2}}}{\Gamma\left(\frac{d}{2}\right)}$$

*(3) If $X \sim \mathcal{N}(\boldsymbol{\mu}, \sigma^2 I_d)$, then*

$$\mathbb{E}_X \|X - \boldsymbol{\mu}\|^p = 2^{\frac{p}{2}} \frac{\Gamma\left(\frac{p+d}{2}\right)}{\Gamma\left(\frac{d}{2}\right)} \sigma^p$$

*Proof.* (1, 2) can be proven by elementary integration. Now we prove (3). By spherical coordinate transformation,

$$\mathbb{E}_X \|X - \boldsymbol{\mu}\|^p = (2\pi\sigma^2)^{-\frac{d}{2}} \int_{u=0}^{\infty} u^{p+d-1} e^{-\frac{u^2}{2\sigma^2}} du \frac{2\pi^{\frac{d}{2}}}{\Gamma\left(\frac{d}{2}\right)} = 2^{\frac{p}{2}} \frac{\Gamma\left(\frac{p+d}{2}\right)}{\Gamma\left(\frac{d}{2}\right)} \sigma^p$$

$\square$

**Lemma A.3** (Gamma tail bound [3])**.** *If $X \sim Gamma(v, c)$, then $P(X > \sqrt{2vt} + ct) \leq e^{-t}$. Or equivalently,*

$$P(X > t) \leq \exp\left(-\frac{v}{c^2}\left(1 + \frac{ct}{v} - \sqrt{1 + \frac{2ct}{v}}\right)\right)$$

*In particular, if $\frac{ct}{v} \geq 4$,*

$$P(X > t) \leq \exp\left(-\frac{v}{c^2}\sqrt{\frac{ct}{v}}\right) = \exp\left(-\sqrt{\frac{vt}{c^3}}\right)$$

**Lemma A.4.** *For $\forall d > 0$, if $r \geq 2\sqrt{d+1}$, then*

$$\int_r^{\infty} u^d e^{-\frac{u^2}{2}} du \leq 2^{\frac{d-1}{2}} \Gamma\left(\frac{d+1}{2}\right) \exp\left(-\frac{r}{2}\sqrt{d+1}\right)$$

*For $p \in \{0, 1, 2\}$, when $r \geq 2\sqrt{d+p}$,*

$$\int_r^{\infty} (u+x)^p u^{d-1} e^{-\frac{u^2}{2}} du \leq 2^{\frac{d}{2}-1} \Gamma\left(\frac{d}{2}\right) (x+d)^p \exp\left(-\frac{r}{2}\sqrt{d}\right)$$

*Proof.* By changing of variables $v = \frac{u^2}{2}$ and integration by parts, we have

$$\int_r^{\infty} u^d e^{-\frac{u^2}{2}} du = 2^{\frac{d-1}{2}} \int_{\frac{r^2}{2}}^{\infty} v^{\frac{d-1}{2}} e^{-v} dv$$

$$= 2^{\frac{d-1}{2}} \Gamma\left(\frac{d+1}{2}\right) P(V > \frac{r^2}{2})$$

where $V \sim Gamma(\frac{d+1}{2}, 1)$. By Lemma A.3, if $r^2 \geq 4(1+d)$,

$$P\left(V > \frac{r^2}{2}\right) \leq \exp\left(-\frac{r}{2}\sqrt{d+1}\right)$$

Hence we have the first inequality. For the second, when $p = 0$, it follows directly from first part. When $p = 1$,

$$\int_r^{\infty} (u+x)^p u^{d-1} e^{-\frac{u^2}{2}} du = \int_r^{\infty} u^d e^{-\frac{u^2}{2}} du + x \int_r^{\infty} u^{d-1} e^{-\frac{u^2}{2}} du$$

$$\leq 2^{\frac{d-1}{2}} \Gamma\left(\frac{d+1}{2}\right) \exp\left(-\frac{r}{2}\sqrt{d+1}\right) + x 2^{\frac{d}{2}-1} \Gamma\left(\frac{d}{2}\right) \exp\left(-\frac{r}{2}\sqrt{d}\right)$$

$$\leq 2^{\frac{d}{2}-1} \Gamma\left(\frac{d}{2}\right) (x+d) \exp\left(-\frac{r}{2}\sqrt{d}\right)$$

where we use $\Gamma\left(\frac{d+1}{2}\right) < \Gamma\left(\frac{d}{2}+1\right) = \frac{d}{2}\Gamma\left(\frac{d}{2}\right)$, and $\exp\left(-\frac{r}{2}\sqrt{d+1}\right) < \exp\left(-\frac{r}{2}\sqrt{d}\right)$ in the last step.

When $p = 2$,

$$\int_r^\infty (u+x)^2 u^{d-1} e^{-\frac{u^2}{2}} du = \int_r^\infty u^{d+1} e^{-\frac{u^2}{2}} du + 2x \int_r^\infty u^d e^{-\frac{u^2}{2}} du$$

$$+ x^2 \int_r^\infty u^{d-1} e^{-\frac{u^2}{2}} du$$

$$\leq 2^{\frac{d}{2}} \Gamma\left(\frac{d}{2}+1\right) \exp\left(-\frac{r}{2}\sqrt{d+2}\right) + 2x \cdot 2^{\frac{d-1}{2}} \Gamma\left(\frac{d+1}{2}\right) \exp\left(-\frac{r}{2}\sqrt{d+1}\right) + x^2 2^{\frac{d}{2}-1} \Gamma\left(\frac{d}{2}\right) \exp\left(-\frac{r}{2}\sqrt{d}\right)$$

$$\leq (d + \sqrt{2}dx + x^2) 2^{\frac{d}{2}-1} \Gamma\left(\frac{d}{2}\right) \exp\left(-\frac{r}{2}\sqrt{d}\right)$$

$$\leq (x+d)^2 2^{\frac{d}{2}-1} \Gamma\left(\frac{d}{2}\right) \exp\left(-\frac{r}{2}\sqrt{d}\right)$$

$\square$

Using Lemma A.4, we can get an easy to use tail bound for Euclidean norm of a Gaussian vector.

**Lemma A.5.** *If $X \sim \mathcal{N}(0, I_d)$, for $r \geq 2\sqrt{d}$, we have*

$$P(\|X\| \geq r) \leq \exp(-\frac{r\sqrt{d}}{2})$$

*Proof.* By spherical coordinate transformation,

$$P(\|X\| \geq r) = \int (2\pi)^{-d/2} \exp(-\|x\|^2/2) dx$$

$$= (2\pi)^{-d/2} \frac{2\pi^{d/2}}{\Gamma\left(\frac{d}{2}\right)} \int_r^\infty r^{d-1} e^{-r^2/2} dr$$

$$\leq \exp\left(-\frac{r}{2}\sqrt{d}\right)$$

$\square$

**Lemma A.6.** *If $X \sim GMM(\pi, \boldsymbol{\mu}^*, \sigma^2 I_d)$, then $X$ is a sub-gaussian random vector with sub-gaussian norm $\sigma + \sum_{i=1}^M \pi_i \|\boldsymbol{\mu}_i^*\|$.*

*Proof.* For any unit vector $u$, consider the random variable $X_u = \langle X, u \rangle$. By the definition in [7], it suffices to show that $X_u$ has a sub-gaussian norm upper bounded by $\sigma + \sum_{i=1}^M \pi_i \|\boldsymbol{\mu}_i^*\|$.

$$\|X_u\|_{\phi_2} = \sup_{p \geq 1} (\mathbb{E}|X_u|^p)^{1/p}$$

For any $p \geq 1$, let $Z$ be the latent variable in the mixture model, we have

$$p^{-1/2} \left(\mathbb{E}|X_u|^p\right)^{1/p} = p^{-1/2} \left(\sum_{i=1}^M \mathbb{E}[|X_u|^p | Z = i] \cdot P(Z = i)\right)^{1/p}$$

$$\leq p^{-1/2} \sum_{i=1}^M \pi_i \left(\mathbb{E}[|X_u|^p | Z = i]\right)^{1/p}$$

$$\overset{(i)}{\leq} p^{-1/2} \sum_{i=1}^M \pi_i \left(\mathbb{E}[|X_u - \boldsymbol{\mu}_i^*|^p | Z = i]^{1/p} + \|\boldsymbol{\mu}_i^*\|\right)$$

$$\leq p^{-1/2} \left(\sum_{i=1}^M \pi_i p^{1/2} \sigma + \|\boldsymbol{\mu}_i^*\|\right) \leq \sigma + \sum_{i=1}^M \pi_i \|\boldsymbol{\mu}_i^*\|$$

where $(i)$ follows from Minkovski's inequality. $\square$

The following lemma characterize the relation between $\|\boldsymbol{\mu}_{\max}^*\|$ and $R_{\max}$.

**Lemma A.7.** *If* $X \sim GMM(\pi, \boldsymbol{\mu}^*, \sigma^2 I_d)$ *with* $\mathbb{E}X = 0$, *let* $\|\boldsymbol{\mu}_{\max}^*\| = \max_i \|\boldsymbol{\mu}_i^*\|$, *then*

$$\|\boldsymbol{\mu}_{\max}^*\| \leq R_{\max} \leq 2\|\boldsymbol{\mu}_{\max}^*\|$$

*Proof.* We first prove $\|\boldsymbol{\mu}_{\max}^*\| \leq R_{\max}$ by contradiction. Assume $\|\boldsymbol{\mu}_{\max}^*\| > R_{\max}$, by definition of $R_{\max}$, all the cluster centers lies in the ball $\mathbb{B}(\|\boldsymbol{\mu}_{\max}^*\|, R_{\max})$, but the origin is outside of the ball, which contradicts the fact that $\mathbb{E}X = \sum_i \pi_i \boldsymbol{\mu}_i^* = 0$.

The second inequality follows from triangle inequality, assume $R_{\max}$ is achieved at $R_{ij}$, then

$$R_{\max} \leq \|\boldsymbol{\mu}_i^*\| + \|\boldsymbol{\mu}_j^*\| \leq 2\|\boldsymbol{\mu}_{\max}^*\|.$$

$\square$

**Lemma A.8.** *A function* $f : \mathbb{R}^n \to \mathbb{R}$ *is* $\sqrt{n}L$ *Lipschitz if there exists a constant* $L$ *such that the restriction of* $f$ *on a certain coordinate is* $L$-*Lipschitz.*

*Proof.* We first relax the norm of difference via a chain of triangle inequalities where each pair of terms only vary on one dimension.

$$|f(x_1, x_2, \cdots, x_n) - f(y_1, y_2, \cdots, x_n)|$$

$$\leq \sum_{i=1}^n |f(y_1, y_2, \cdots, y_{i-1}, x_i, x_{i+1}, \cdots, x_n) - f(y_1, y_2, \cdots, y_{i-1}, y_i, x_{i+1}, \cdots, x_n)|$$

$$\leq \sum_{i=1}^n L|x_i - y_i| \leq \sqrt{n}L \|x - y\|$$

$\square$

# B  Proofs in Section 4

*Proof of Lemma 1.* By (2), $\nabla_{\boldsymbol{\mu}_i} q(\boldsymbol{\mu}) = \mathbb{E}_X w_i(X; \boldsymbol{\mu}^*)(X - \boldsymbol{\mu}_i)$. Without loss of generality, we only show the claim for $i = 1$. That is equivalent of saying, if $X \sim \text{GMM}(\pi, \boldsymbol{\mu}^*)$, we have $\mathbb{E}[w_1(X; \boldsymbol{\mu}^*)(X - \boldsymbol{\mu}_1^*)] = 0$. Denote $\mathcal{N}(\boldsymbol{\mu}_i^*, \Sigma)$ as $\mathcal{N}_i$ and its distribution as $\phi_i(X)$. Decompose the left hand side with respect to the mixture components, we have

$$\mathbb{E}[w_1(X)X] = \sum_i \pi_i \mathbb{E}_{X \sim \mathcal{N}_i}[w_1(X)X]$$

$$= \sum_i \pi_i \int \phi_i(X) \frac{\pi_1 \phi_1(X)}{\sum_k \pi_k \phi_k(X)} X dx$$

$$= \pi_1 \mathbb{E}_{X \sim \mathcal{N}_1} X = \pi_1 \boldsymbol{\mu}_1^*$$

Similarly $\mathbb{E}[w_1(X)] = \pi_1$. Hence $\nabla_{\boldsymbol{\mu}_1} q(\boldsymbol{\mu}) = \mathbb{E}_X w_1(X; \boldsymbol{\mu}^*)(X - \boldsymbol{\mu}_1) = \pi_1(\boldsymbol{\mu}_1^* - \boldsymbol{\mu}_1)$. This completes the proof. $\square$

*Proof of Theorem 3.* Define By Lemma 1, the GS condition is equivalent to

$$\left\|\nabla Q(\boldsymbol{\mu}|\boldsymbol{\mu}^t) - \nabla q(\boldsymbol{\mu})\right\| \leq \gamma \|\boldsymbol{\mu}^t - \boldsymbol{\mu}^*\|$$

By triangle inequality,

$$\begin{aligned}\|\boldsymbol{\mu}_1^{t+1} - \boldsymbol{\mu}_1^*\| &= \left\|\boldsymbol{\mu}_1^t - \boldsymbol{\mu}_1^* + s\nabla Q(\boldsymbol{\mu}|\boldsymbol{\mu}^t)\right\| \\ &\leq \left\|\boldsymbol{\mu}_1^t - \boldsymbol{\mu}_1^* + s\nabla q(\boldsymbol{\mu})\right\| + s\left\|\nabla Q(\boldsymbol{\mu}|\boldsymbol{\mu}^t) - \nabla q(\boldsymbol{\mu})\right\| \\ &\leq \frac{\pi_{\max} - \pi_{\min}}{\pi_{\max} + \pi_{\min}} \left\|\boldsymbol{\mu}_1^t - \boldsymbol{\mu}_1^*\right\| + \frac{2}{\pi_{\max} + \pi_{\min}} \gamma \left\|\boldsymbol{\mu}_1^t - \boldsymbol{\mu}_1^*\right\| \\ &\leq \frac{\pi_{\max} - \pi_{\min} + 2\gamma}{\pi_{\max} + \pi_{\min}} \left\|\boldsymbol{\mu}_1^t - \boldsymbol{\mu}_1^*\right\|\end{aligned}$$

To see why the last inequality hold, notice that $q(\boldsymbol{\mu})$ has largest eigenvalue $-\pi_{\min}$ and smallest eigenvalue $-\pi_{\max}$. Apply the classical result for gradient descent, with step size $s = \frac{2}{\pi_{\max}+\pi_{\min}}$ guarantees

$$\left\|\boldsymbol{\mu}_1^t - \boldsymbol{\mu}_1^* + s\nabla q(\boldsymbol{\mu})\right\| \le \frac{\pi_{\max} - \pi_{\min}}{\pi_{\max} + \pi_{\min}} \left\|\boldsymbol{\mu}_1^t - \boldsymbol{\mu}_1^*\right\|$$

$\square$

## B.1 Proofs of Theorem 4

We start with two lemmas.

**Lemma B.1.** *For $X \sim GMM(\pi, \boldsymbol{\mu}^*, I_d)$, if $R_{\min} = \tilde{\Omega}(\sqrt{d})$, and $\boldsymbol{\mu}_i \in \mathbb{B}(\boldsymbol{\mu}_i^*, a), \forall i \in [M]$ where*

$$a \le \frac{R_{\min}}{2} - \sqrt{d}\max(4\sqrt{2[\log(R_{\min}/4)]_+}, 8\sqrt{3}).$$

*Then for $p = 0, 1, 2$ and $\forall i \in [M]$, we have*

$$\mathbb{E}_X w_i(X; \boldsymbol{\mu})(1 - w_i(X; \boldsymbol{\mu}))\|X - \boldsymbol{\mu}_i\|^p \le 2M\left(\frac{3}{2}R_{\max} + d\right)^p \exp\left(-\left(\frac{R_{\min}}{2} - a\right)^2 \sqrt{d}/8\right)$$

Using the same techniques, for the cross terms, we have the following lemma.

**Lemma B.2.** *Assume $X \sim GMM(\pi, \boldsymbol{\mu}^*, I_d)$, and $\boldsymbol{\mu}_i \in \mathbb{B}(\boldsymbol{\mu}_i^*, a), \forall i \in [M]$. Under the same conditions as in Lemma B.1, we have for $\forall i \ne j \in [M]$,*

$$\mathbb{E}_X[w_i(X; \boldsymbol{\mu})w_j(X; \boldsymbol{\mu})\|X - \boldsymbol{\mu}_i\| \cdot \|X - \boldsymbol{\mu}_j\|] \le (1 + 2\kappa)\left(\frac{3}{2}R_{\max} + d\right)^2 \exp\left(-\left(\frac{R_{\min}}{2} - a\right)^2 \sqrt{d}/8\right)$$

*Proof of Lemma B.1.* Without loss of generality, we prove the claim for $i = 1$. Recall the definition of $w_i(X; \boldsymbol{\mu})$ from Equation 1. For $p \in \{0, 1, 2\}$,

$$\begin{aligned}
&\mathbb{E}_X w_1(X; \boldsymbol{\mu})(1 - w_1(X; \boldsymbol{\mu}))\|X - \boldsymbol{\mu}_1\|^p \\
&= \sum_{i \in [M]} \pi_i \mathbb{E}_{X \sim \mathcal{N}(\boldsymbol{\mu}_i^*)} w_1(X; \boldsymbol{\mu})(1 - w_1(X; \boldsymbol{\mu}))\|X - \boldsymbol{\mu}_1\|^p \\
&\le \pi_1 \mathbb{E}_{X \sim \mathcal{N}(\boldsymbol{\mu}_1^*)} w_1(X; \boldsymbol{\mu})(1 - w_1(X; \boldsymbol{\mu}))\|X - \boldsymbol{\mu}_1\|^p + \sum_{i \ne 1} \pi_i \mathbb{E}_{X \sim \mathcal{N}(\boldsymbol{\mu}_i^*)} w_1(X; \boldsymbol{\mu})\|X - \boldsymbol{\mu}_1\|^p
\end{aligned}$$

(B.1)

First let us look at the first term. Define event $\mathcal{E}_r^{(1)} = \{X : X \sim \mathcal{N}(\boldsymbol{\mu}_1^*); \|X - \boldsymbol{\mu}_1^*\| \le r\}$ for some $r > 0$. We will see later that we need $r < \frac{R_{\min}}{2} - a$. Then for $X \in \mathcal{E}_r^{(1)}$ using triangle inequality, we have

$$\|X - \boldsymbol{\mu}_i\| \begin{cases} \le \|X - \boldsymbol{\mu}_i^*\| + \|\boldsymbol{\mu}_i^* - \boldsymbol{\mu}_i\| \le r + a & i = 1 \\ \ge \|\boldsymbol{\mu}_i - \boldsymbol{\mu}_1^*\| - \|X - \boldsymbol{\mu}_1^*\| \ge \|\boldsymbol{\mu}_i^* - \boldsymbol{\mu}_1^*\| - \|\boldsymbol{\mu}_i^* - \boldsymbol{\mu}_i\| - r \ge R_{\min} - r - a & i \ne 1 \end{cases}$$

(B.2)

$$\begin{aligned}
&\mathbb{E}_{X \sim \mathcal{N}(\boldsymbol{\mu}_1^*)} w_1(X; \boldsymbol{\mu})(1 - w_1(X; \boldsymbol{\mu}))\|X - \boldsymbol{\mu}_1\|^p \\
&= \mathbb{E}[w_1(X; \boldsymbol{\mu})(1 - w_1(X; \boldsymbol{\mu}))\|X - \boldsymbol{\mu}_1\|^p | \mathcal{E}_r^{(1)}]P(\mathcal{E}_r^{(1)}) \\
&\quad + \mathbb{E}[w_1(X; \boldsymbol{\mu})(1 - w_1(X; \boldsymbol{\mu}))\|X - \boldsymbol{\mu}_1\|^p | \mathcal{E}_r^{(1)c}]P(\mathcal{E}_r^{(1)c})
\end{aligned}$$

In view of the fact that $w_1(X; \boldsymbol{\mu})$ is monotonically decreasing w.r.t. $\|X - \boldsymbol{\mu}_i\|$ and increasing w.r.t. $\|X - \boldsymbol{\mu}_1\|$, we have

$$1 - w_1(X; \boldsymbol{\mu}) \leq \frac{(1 - \pi_1) \exp\left(-\frac{(R_{\min} - r - a)^2}{2}\right)}{\pi_1 \exp\left(-\frac{(r + a)^2}{2}\right) + (1 - \pi_1) \exp\left(-\frac{(R_{\min} - r - a)^2}{2}\right)}$$

$$\leq \frac{1 - \pi_1}{\pi_1} \exp\left(-\frac{1}{2} R_{\min}(R_{\min} - 2r - 2a)\right)$$

Also notice that $w_1(X; \boldsymbol{\mu}) \leq 1$, we have

$$\mathbb{E}[w_1(X; \boldsymbol{\mu})(1 - w_1(X; \boldsymbol{\mu}))\|X - \boldsymbol{\mu}_1\|^p | \mathcal{E}_r^{(1)}] P(\mathcal{E}_r^{(1)})$$

$$\leq \frac{1 - \pi_1}{\pi_1} \exp\left(-\frac{1}{2} R_{\min}(R_{\min} - 2r - 2a)\right)(r + a)^p$$

For $\mathcal{E}_r^{(1)c}$, note $w_1(X; \boldsymbol{\mu})(1 - w_1(X; \boldsymbol{\mu})) \leq \frac{1}{4}$, we have for $p = 1$,

$$\mathbb{E}[w_1(X; \boldsymbol{\mu})(1 - w_1(X; \boldsymbol{\mu}))\|X - \boldsymbol{\mu}_1\| | \mathcal{E}_r^{(1)c}] P(\mathcal{E}_r^{(1)c})$$

$$\leq \frac{1}{4} \int_{u=r}^{\infty} (u + a)(2\pi)^{-\frac{d}{2}} \exp\left(-\frac{u^2}{2}\right) \cdot \frac{2\pi^{\frac{d}{2}}}{\Gamma\left(\frac{d}{2}\right)} u^{d-1} du$$

$$\leq \frac{1}{4}(2\pi)^{-\frac{d}{2}} \frac{2\pi^{\frac{d}{2}}}{\Gamma\left(\frac{d}{2}\right)} \int_{u=r}^{\infty} (u + a) \exp\left(-\frac{u^2}{2}\right) u^{d-1} du$$

$$\overset{(i)}{\leq} \frac{a + d}{4} \exp\left(-\frac{r}{2}\sqrt{d}\right)$$

The inequality (i) follows from Lemma A.4 when $r > 2\sqrt{d+1}$. Similarly, for $p = 2$,

$$\mathbb{E}[w_1(X; \boldsymbol{\mu})(1 - w_1(X; \boldsymbol{\mu}))\|X - \boldsymbol{\mu}_1\|^2 | \mathcal{E}_r^{(1)c}] P(\mathcal{E}_r^{(1)c})$$

$$\leq \frac{2^{-\frac{d}{2}-1}}{\Gamma\left(\frac{d}{2}\right)} \int_r^{\infty} (u + a)^2 u^{d-1} e^{-\frac{u^2}{2}} du \overset{(ii)}{\leq} \frac{(a + d)^2}{4} \exp\left(-\frac{r}{2}\sqrt{d}\right)$$

The inequality (ii) follows from Lemma A.4 when $r > 2\sqrt{d+1}$ and $p = 2$. Therefore for the first mixture we have,

$$\pi_1 \mathbb{E}_{X \sim \mathcal{N}(\boldsymbol{\mu}_1^*)} w_1(X; \boldsymbol{\mu})(1 - w_1(X; \boldsymbol{\mu}))\|X - \boldsymbol{\mu}_1\|^p$$

$$\leq (1 - \pi_1)(r + a)^p \exp\left(-\frac{1}{2} R_{\min}(R_{\min} - 2r - 2a)\right) + \pi_1 \frac{(a + d)^p}{4} \exp\left(-\frac{r}{2}\sqrt{d}\right) \quad \text{(B.3)}$$

Next we bound $\mathbb{E}_{X \sim \mathcal{N}(\boldsymbol{\mu}_i^*)} w_1(X; \boldsymbol{\mu})\|X - \boldsymbol{\mu}_1\|^p$ for $i \neq 1$. For some $0 < r < \frac{R}{2} - a$, we have

$$\pi_i \mathbb{E}_{X \sim \mathcal{N}(\boldsymbol{\mu}_i^*)} w_1(X; \boldsymbol{\mu})\|X - \boldsymbol{\mu}_1\|^p$$

$$= \int_X \frac{\pi_1 \phi(X; \boldsymbol{\mu}_1) \cdot \pi_i \phi(X; \boldsymbol{\mu}_i^*)}{\sum_j \pi_j \phi(X; \boldsymbol{\mu}_j)} \|X - \boldsymbol{\mu}_1\|^p dX$$

$$= \underbrace{\int_{X \in \mathbb{B}(\boldsymbol{\mu}_i^*, r)} \frac{\pi_1 \phi(X; \boldsymbol{\mu}_1) \cdot \pi_i \phi(X; \boldsymbol{\mu}_i^*)}{\sum_j \pi_j \phi(X; \boldsymbol{\mu}_j)} \|X - \boldsymbol{\mu}_1\|^p dX}_{I_1^{(p)}} + \underbrace{\int_{X \notin \mathbb{B}(\boldsymbol{\mu}_i^*, r)} \frac{\pi_1 \phi(X; \boldsymbol{\mu}_1) \cdot \pi_i \phi(X; \boldsymbol{\mu}_i^*)}{\sum_j \pi_j \phi(X; \boldsymbol{\mu}_j)} \|X - \boldsymbol{\mu}_1\|^p dX}_{I_2^{(p)}}$$

$$\text{(B.4)}$$

When $\|X - \boldsymbol{\mu}_i^*\| \leq r$, since by assumption $\|\boldsymbol{\mu}_i - \boldsymbol{\mu}_i^*\| \leq a$,

$$\frac{\phi(X; \boldsymbol{\mu}_i^*)}{\phi(X; \boldsymbol{\mu}_i)} = \exp\left(\frac{\|X - \boldsymbol{\mu}_i\|^2}{2} - \frac{\|X - \boldsymbol{\mu}_i^*\|^2}{2}\right)$$

$$= \exp\left(\left(X - \frac{\boldsymbol{\mu}_i + \boldsymbol{\mu}_i^*}{2}\right)^T (\boldsymbol{\mu}_i - \boldsymbol{\mu}_i^*)\right)$$

$$\text{(B.5)}$$

Since by Cauchy-Schwarz we have $|(X - \frac{\mu_i + \mu_i^*}{2})^T(\mu_i - \mu_i^*)| = |(X - \mu_i^* + \frac{\mu_i^* - \mu_i}{2})^T(\mu_i - \mu_i^*)| \le (r + a/2)a$, we have:

$$\exp\left(-(r + \frac{a}{2})a\right) \le \frac{\phi(X; \mu_i^*)}{\phi(X; \mu_i)} \le \exp\left((r + \frac{a}{2})a\right) \tag{B.6}$$

For such $X$, $\phi(X; \mu_1) \le (2\pi)^{-\frac{d}{2}} \exp\left(-\frac{(R_{\min} - r - a)^2}{2}\right)$, and we have

$$
\begin{aligned}
I_1^{(p)} &= \int_{X \in \mathbb{B}(\mu_i^*, r)} \frac{\pi_1 \phi(X; \mu_1) \pi_i \phi(X; \mu_i^*)}{\sum_j \pi_j \phi(X; \mu_j)} \|X - \mu_1\|^p dX \\
&\le \int_{X \in \mathbb{B}(\mu_i^*, r)} \frac{\pi_1 \phi(X; \mu_1) \pi_i \phi(X; \mu_i) \exp\left((r + \frac{a}{2})a\right)}{\sum_j \pi_j \phi(X; \mu_j)} \|X - \mu_1\|^p dX \\
&\le \pi_1 \exp\left((r + \frac{a}{2})a\right) \int_{X \in \mathbb{B}(\mu_i^*, r)} \phi(X; \mu_1)\|X - \mu_1\|^p dX \\
&\le \pi_1 (2\pi)^{-d/2} \exp\left((r + \frac{a}{2})a\right) (R_{\max} + a + r)^p \exp\left(-\frac{(R_{\min} - r - a)^2}{2}\right) \frac{\pi^{d/2}}{\Gamma(\frac{d}{2} + 1)} r^d \\
&\le \frac{\pi_1 2^{-d/2}}{\Gamma(\frac{d}{2} + 1)} \exp\left((r + \frac{a}{2})a - \frac{(R_{\min} - r - a)^2}{2}\right) (R_{\max} + a + r)^p r^d \\
&\le \pi_1 2^{1-d} \exp\left(R_{\min}\left(a - \frac{R_{\min}}{2}(1 - r/R_{\min})^2\right)\right) (R_{\max} + a + r)^p r^d
\end{aligned}
$$

The last inequality follows from the fact that $\Gamma\left(\frac{d}{2} + 1\right) \ge ([\frac{d}{2}])! \ge 2^{\frac{d}{2} - 1}$. On the other hand, for $I_2$, since $w_1(X; \mu) \le 1$, taking spherical coordinate transformation we have,

$$
\begin{aligned}
I_2^{(p)} &\le \int_{\|X - \mu_i^*\| \ge r} \pi_i \phi(X; \mu_i^*)\|X - \mu_1\|^p dX \\
&\le \pi_i \int_{\|X - \mu_i^*\| \ge r} (2\pi)^{-d/2} \exp(-\frac{\|X - \mu_i^*\|^2}{2})\|X - \mu_1\|^p dX \\
&\le \frac{\pi_i 2^{1-d/2}}{\Gamma(\frac{d}{2})} \int_{u=r}^{\infty} u^{d-1} \exp\left(-\frac{u^2}{2}\right) (u + R_{\max} + a)^p du
\end{aligned}
$$

Apply Lemma A.4, when $r \ge 2\sqrt{d+2}$, for $p \in \{0, 1, 2\}$

$$I_2^{(p)} \le \pi_i (R_{\max} + a + d)^p \exp\left(-\frac{r}{2}\sqrt{d}\right) \tag{B.7}$$

Summing up $I_1$ and $I_2$, for any $0 < r < R_{\min}/2$, from (B.4) we get:

$$
\begin{aligned}
&\pi_i \mathbb{E}_{X \sim \mathcal{N}(\mu_i^*)} w_1(X; \mu)\|X - \mu_1\|^p \\
&\le \pi_1 2^{1-d} \exp\left(R_{\min}\left(a - \frac{R_{\min}}{2}(1 - r/R_{\min})^2\right)\right) (R_{\max} + a + r)^p r^d + \pi_i (R_{\max} + a + d)^p \exp\left(-\frac{r}{2}\sqrt{d}\right)
\end{aligned}
\tag{B.8}
$$

Now plugging Eq. (B.3) and Eq. (B.8) into Eq. (B.1) gives,

$$\mathbb{E}_X w_1(X;\boldsymbol{\mu})(1-w_1(X;\boldsymbol{\mu}))\|X-\boldsymbol{\mu}_1\|^p$$

$$\leq (1-\pi_1)(r+a)^p \exp\left(-\frac{1}{2}R_{\min}(R_{\min}-2r-2a)\right) + \pi_1\frac{(a+d)^p}{4}\exp\left(-\frac{r}{2}\sqrt{d}\right)$$

$$+ \pi_1(M-1)2^{1-d}\exp\left(R_{\min}\left(a-\frac{R_{\min}}{2}(1-r/R_{\min})^2\right)\right)(R_{\max}+a+r)^p r^d$$

$$+ (1-\pi_1)(R_{\max}+a+d)^p \exp\left(-\frac{r}{2}\sqrt{d}\right)$$

$$\leq \underbrace{(1-\pi_1)(r+a)^p \exp\left(-\frac{1}{2}R_{\min}(R_{\min}-2r-2a)\right)}_{(A)} + \underbrace{(R_{\max}+a+d)^p \exp\left(-\frac{r}{2}\sqrt{d}\right)}_{(B)}$$

$$+ \underbrace{2\pi_1(M-1)\exp\left(R_{\min}\left(a-\frac{R_{\min}}{2}(1-r/R_{\min})^2\right)+d\log(r/2)\right)(R_{\max}+a+r)^p}_{(C)}$$

Note that in order to have a negative term inside exponential of (A), we require $r+a < \frac{R_{\min}}{2}$. In order to ensure the same for (C), we need:

$$a < \frac{R_{\min}}{2}\left(1-\frac{r}{R_{\min}}\right)^2 \tag{B.9}$$

If $r^2 \geq 2d\log(r/2)$, then we have:

$$\exp\left(R_{\min}\left(a-\frac{R_{\min}}{2}(1-r/R_{\min})^2\right)+d\log(r/2)\right) \leq \exp\left(R_{\min}\left(a-\frac{R_{\min}}{2}(1-r/R_{\min})^2\right)+r^2/2\right)$$

$$\leq \exp\left(R_{\min}a - \left(\frac{R_{\min}^2}{2}-rR_{\min}+\frac{r^2}{2}\right)+\frac{r^2}{2}\right)$$

$$= \exp\left(-\frac{1}{2}R_{\min}(R_{\min}-2r-2a)\right)$$

Therefore, $(A)+(C) \leq (1-\pi_1+2\pi_1(M-1))(R_{\max}+a+r)^p \exp\left(-\frac{1}{2}R_{\min}(R_{\min}-2r-2a)\right)$

Finally, if $r \leq R_{\min}\frac{R_{\min}/2-a}{R_{\min}+\sqrt{d}/2}$, we have:

$$\exp\left(-\frac{1}{2}R_{\min}(R_{\min}-2r-2a)\right) \leq \exp(-\frac{r}{2}\sqrt{d})$$

Hence,

$$(A)+(B)+(C) \leq (2-\pi_1+2\pi_1(M-1))\left(\frac{3}{2}R_{\max}+d\right)^p \exp\left(-\frac{r}{2}\sqrt{d}\right)$$

$$\leq 2M\left(\frac{3}{2}R_{\max}+d\right)^p \exp\left(-\frac{r}{2}\sqrt{d}\right)$$

Set

$$r = \frac{R_{\min}/2-a}{4}, \quad a \leq \frac{R_{\min}}{2} \tag{B.10}$$

then Eq (B.9) and $a+r \leq \frac{R_{\min}}{2}$ are automatically satisfied. When $R_{\min} \geq \frac{\sqrt{d}}{6}$, we have $r \leq R_{\min}\frac{R_{\min}/2-a}{R_{\min}+\sqrt{d}/2}$. Finally in order to meet the constraints

$$r \geq 2\sqrt{d+2} \Leftarrow r \geq 3\sqrt{d} \tag{B.11}$$

$$r^2 \geq 2d\log r/2 \tag{B.12}$$

we need

$$\frac{R_{\min}/2 - a}{4} \geq \max(\sqrt{2d[\log(R_{\min}/4)]_+}, 2\sqrt{3}\sqrt{d})$$

$$a \leq \frac{R_{\min}}{2} - \sqrt{d}\max(4\sqrt{2[\log(R_{\min}/4)]_+}, 8\sqrt{3})$$

The right hand side of last inequality is non-negative when $R_{\min} = \tilde{\Omega}(\sqrt{d})$. Under these conditions, with Eq. (B.10) plugged in, we have

$$\mathbb{E}_X w_1(X; \boldsymbol{\mu})(1 - w_1(X; \boldsymbol{\mu}))\|X - \boldsymbol{\mu}_1\|^p \leq 2M\left(\frac{3}{2}R_{\max} + d\right)^p \exp\left(-\left(\frac{R_{\min}}{2} - a\right)^2 \sqrt{d}/8\right)$$

$\square$

*Proof of Lemma B.2.* For any $r \leq \frac{R_{\min}}{2} - a$, define $\mathcal{E}_0 = \{X : \exists i, \text{ such that } Z_X = i, \|X - \boldsymbol{\mu}_i^*\| > r\}$ and $\mathcal{E}_k = \{X : Z_X = k, \|X - \boldsymbol{\mu}_k^*\| \leq r\}$.

$$\mathbb{E}_X\left[w_i(X; \boldsymbol{\mu})w_j(X; \boldsymbol{\mu})\|X - \boldsymbol{\mu}_i\| \cdot \|X - \boldsymbol{\mu}_j\|\right]$$
$$\leq \underbrace{\mathbb{E}_X\left[w_i(X; \boldsymbol{\mu})w_j(X; \boldsymbol{\mu})\|X - \boldsymbol{\mu}_i\|\|X - \boldsymbol{\mu}_j\||\mathcal{E}_0\right]P(\mathcal{E}_0)}_{I_0}$$
$$+ \sum_{k \in [M]} \pi_k \underbrace{\mathbb{E}_{X \sim \mathcal{N}(\boldsymbol{\mu}_k^*)}\left[w_i(X; \boldsymbol{\mu})w_j(X; \boldsymbol{\mu})\|X - \boldsymbol{\mu}_i\|\|X - \boldsymbol{\mu}_j\|\|X - \boldsymbol{\mu}_k\| \leq r\right]}_{I_k}$$

First we look at $I_0$, this again can be decomposed as the sum over mixtures. Similarly as in Eq. (B.7), we have

$$I_0 \leq (R_{\max} + a + d)^2 \exp\left(-\frac{r}{2}\sqrt{d}\right)$$

For $I_k$, by Eq. (B.6),

$$I_k = \int_X \frac{\pi_i \phi(X; \boldsymbol{\mu}_i)\pi_j \phi(X; \boldsymbol{\mu}_j)\pi_k \phi(X; \boldsymbol{\mu}_k^*)}{(\sum_\ell \pi_\ell \phi(X; \boldsymbol{\mu}_t))^2}\|X - \boldsymbol{\mu}_i\| \cdot \|X - \boldsymbol{\mu}_j\|dX$$
$$\leq \int_X \frac{\pi_i \phi(X; \boldsymbol{\mu}_i)\pi_j \phi(X; \boldsymbol{\mu}_j)\pi_k \phi(X; \boldsymbol{\mu}_k)\exp((r + a/2)a)}{(\sum_\ell \pi_\ell \phi(X; \boldsymbol{\mu}_\ell))^2}\|X - \boldsymbol{\mu}_i\| \cdot \|X - \boldsymbol{\mu}_j\|dX$$
$$\leq \kappa\pi_k 2\pi^{-\frac{d}{2}}\exp\left(-\frac{R_{(\min - r - a)}^2}{2}\right)\exp((r + a/2)a)(R_{\max} + r + a)^2\frac{\pi^{d/2}}{\Gamma\left(\frac{d}{2} + 1\right)}r^d \quad \text{(B.13)}$$
$$\leq \pi_k \kappa 2^{-d/2}\frac{1}{\Gamma\left(\frac{d}{2} + 1\right)}r^d \exp\left((r + a/2)a - \frac{(R_{\min} - r - a)^2}{2}\right)(R_{\max} + r + a)^2$$
$$\leq 2\pi_k \kappa \exp\left(R_{\min}\left(a - \frac{R_{\min}}{2}\left(1 - \frac{r}{R_{\min}}\right)^2\right) + d\log(r/2)\right)(R_{\max} + r + a)^2$$

Adding up $I_k$'s and $I_0$, we have

$$\mathbb{E}_X\left[w_i(X; \boldsymbol{\mu})w_j(X; \boldsymbol{\mu})\|X - \boldsymbol{\mu}_i\|\|X - \boldsymbol{\mu}_j\|\right]$$
$$\leq (R_{\max} + a + d)^2 \exp\left(-\frac{r}{2}\sqrt{d}\right)$$
$$+ 2\kappa\exp\left(R_{\min}\left(a - \frac{R_{\min}}{2}\left(1 - \frac{r}{R_{\min}}\right)^2\right) + d\log(r/2)\right)(R_{\max} + r + a)^2$$

Take $r = \frac{1}{4}\left(\frac{R_{\min}}{2} - a\right)$, we have $R_{\min}\left(a - \frac{R_{\min}}{2}\left(1 - \frac{r}{R_{\min}}\right)^2\right) + d\log(r/2) \leq -\frac{r}{2}\sqrt{d}$. Therefore,

$$\mathbb{E}_X[w_i(X; \boldsymbol{\mu})w_j(X; \boldsymbol{\mu})\|X - \boldsymbol{\mu}_i\| \cdot \|X - \boldsymbol{\mu}_j\|]$$
$$\leq (1 + 2\kappa)\left(\frac{3}{2}R_{\max} + d\right)^2 \exp\left(-\left(\frac{R_{\min}}{2} - a\right)^2 \sqrt{d}/8\right)$$

$\square$

*Proof of Theorem 4.* Consider the difference of the gradient corresponding to $\boldsymbol{\mu}_i$, without loss of generality, assume $i = 1$.

$$\nabla_{\boldsymbol{\mu}_1} Q(\boldsymbol{\mu}^t|\boldsymbol{\mu}^t) - \nabla q(\boldsymbol{\mu}^t) = \mathbb{E}(w_1(X;\boldsymbol{\mu}^t) - w_1(X;\boldsymbol{\mu}^*))(X - \boldsymbol{\mu}_1^t) \tag{B.14}$$

For any given $X$, consider the function $\boldsymbol{\mu} \to w_1(X;\boldsymbol{\mu})$, we have

$$\nabla_{\boldsymbol{\mu}} w_1(X;\boldsymbol{\mu}) = \begin{pmatrix} w_1(X;\boldsymbol{\mu})(1 - w_1(X;\boldsymbol{\mu}))(X - \boldsymbol{\mu}_1)^T \\ -w_1(X;\boldsymbol{\mu})w_2(X;\boldsymbol{\mu})(X - \boldsymbol{\mu}_2)^T \\ \vdots \\ -w_1(X;\boldsymbol{\mu})w_M(X;\boldsymbol{\mu})(X - \boldsymbol{\mu}_M)^T \end{pmatrix} \tag{B.15}$$

Let $\boldsymbol{\mu}^u = \boldsymbol{\mu}^* + u(\boldsymbol{\mu}^t - \boldsymbol{\mu}^*), \forall u \in [0,1]$, obviously $\boldsymbol{\mu}^u \in \otimes_{i=1}^M \mathbb{B}(\boldsymbol{\mu}_i^*, \|\boldsymbol{\mu}_i^t - \boldsymbol{\mu}_i^*\|) \subset \otimes_{i=1}^M \mathbb{B}(\boldsymbol{\mu}_i^*, a)$. By Taylor's theorem,

$$\|\mathbb{E}(w_1(X;\boldsymbol{\mu}_1^t) - w_1(X;\boldsymbol{\mu}_1^*))(X - \boldsymbol{\mu}_1^t)\| = \left\| \mathbb{E}\left[ \int_{u=0}^1 \nabla_u w_1(X;\boldsymbol{\mu}^u) du (X - \boldsymbol{\mu}_1^t) \right] \right\|$$

$$= \left\| \int_{u=0}^1 \mathbb{E} w_1(X;\boldsymbol{\mu}^u)(1 - w_1(X;\boldsymbol{\mu}^u))(X - \boldsymbol{\mu}_1^u)^T (\boldsymbol{\mu}_1^t - \boldsymbol{\mu}_1^*)(X - \boldsymbol{\mu}_1^t) du \right.$$

$$\left. - \sum_{i \neq 1} \int_{u=0}^1 \mathbb{E} w_1(X;\boldsymbol{\mu}^u)w_i(X;\boldsymbol{\mu}^u))(X - \boldsymbol{\mu}_2^u)^T (\boldsymbol{\mu}_2^t - \boldsymbol{\mu}_2^*)(X - \boldsymbol{\mu}_1^t) du \right\| \tag{B.16}$$

$$\leq U_1 \|\boldsymbol{\mu}_1^t - \boldsymbol{\mu}_1^*\|_2 + \sum_{i \neq 1} U_i \|\boldsymbol{\mu}_i^t - \boldsymbol{\mu}_i^*\|_2$$

where

$$U_1 = \sup_{u \in [0,1]} \|\mathbb{E} w_1(X;\boldsymbol{\mu}^u)(1 - w_1(X;\boldsymbol{\mu}^u))(X - \boldsymbol{\mu}_1^t)(X - \boldsymbol{\mu}_1^u)^T\|_{op}$$

$$U_i = \sup_{u \in [0,1]} \|\mathbb{E} w_1(X;\boldsymbol{\mu}^u)w_i(X;\boldsymbol{\mu}^u)(X - \boldsymbol{\mu}_1^t)(X - \boldsymbol{\mu}_2^u)^T\|_{op}$$

For $U_1$ by triangle inequality we have,

$$U_1 \leq \sup_{u \in [0,1]} \|\mathbb{E} w_1(X;\boldsymbol{\mu}^u)(1 - w_1(X;\boldsymbol{\mu}^u))(X - \boldsymbol{\mu}_1^u)(X - \boldsymbol{\mu}_1^u)^T\|_{op}$$

$$+ \sup_{u \in [0,1]} \|\mathbb{E} w_1(X;\boldsymbol{\mu}^u)(1 - w_1(X;\boldsymbol{\mu}^u))(\boldsymbol{\mu}_1^u - \boldsymbol{\mu}_1^t)(X - \boldsymbol{\mu}_1^u)^T\|_{op}$$

$$\leq \sup_{u \in [0,1]} \|\mathbb{E} w_1(X;\boldsymbol{\mu}^u)(1 - w_1(X;\boldsymbol{\mu}^u))(X - \boldsymbol{\mu}_1^u)(X - \boldsymbol{\mu}_1^u)^T\|_{op}$$

$$+ a \sup_{u \in [0,1]} \|\mathbb{E} w_1(X;\boldsymbol{\mu}^u)(1 - w_1(X;\boldsymbol{\mu}^u))(X - \boldsymbol{\mu}_1^u)\| \tag{B.17}$$

We now develop an uniform bound for the operator norm. For any $u \in [0,1]$, there exists a rotation matrix $O$, such that all $R\boldsymbol{\mu}_i^u, i \in [M]$ have non-zero entries in the leading $\min\{d, M\}$ coordinates, and zeros for the remaining $[d - M]_+$ coordinates. Denote $\tilde{X} := OX$, then $\tilde{X}|Z = i \sim \mathcal{N}(O\boldsymbol{\mu}_i^*, I_d)$. Let

$$O\boldsymbol{\mu}_i^u = [\tilde{\boldsymbol{\mu}}_i^u, 0_{[d-M]_+}] \text{ and } O\boldsymbol{\mu}_i^* = [v_i^{\min\{d,M\}}, v_i^{[d-M]_+}], \quad \tilde{\boldsymbol{\mu}}_i^u \in \mathbb{R}^{\min\{d,M\}}$$

For ease of notation, we assume $d \geq M$ for now, the other case can be derived without much modification. We can rewrite

$$(X - \boldsymbol{\mu}_1^u)(X - \boldsymbol{\mu}_1^u)^T = O^T \begin{bmatrix} (\tilde{X}^M - \tilde{\boldsymbol{\mu}}_1^u)(\tilde{X}^M - \tilde{\boldsymbol{\mu}}_1^u)^T & (\tilde{X}^M - \tilde{\boldsymbol{\mu}}_1^u)(\tilde{X}^{d-M})^T \\ (\tilde{X}^{d-M})(\tilde{X}^M - \tilde{\boldsymbol{\mu}}_1^u)^T & (\tilde{X}^{d-M})(\tilde{X}^{d-M})^T \end{bmatrix} O$$

Note by the rotation, $w_i(X;\boldsymbol{\mu})$ only depend on the first $M$ coordinates. And by isotropicity, $\tilde{X}^M$ and $\tilde{X}^{d-M}$ are independent. By $\mathbb{E}\tilde{X}^{d-M} = 0$ (since we assume that the centroid of the means is at zero, and a rotation does not change that) and $\mathbb{E}\tilde{X}^{d-M}(\tilde{X}^{d-M})^T = I_{d-M} + \sum_i \pi_i (v_i^{d-M})(v_i^{d-M})^T$, we have,

$$\|\mathbb{E}w_1(X;\boldsymbol{\mu}^u)(1-w_1(X;\boldsymbol{\mu}^u))(X-\boldsymbol{\mu}_1^u)(X-\boldsymbol{\mu}_1^u)^T\|_{op} = \left\|\begin{bmatrix} D_1 & 0 \\ 0 & D_2 \end{bmatrix}\right\|_{op}$$

$$\leq \max\{\|D_1\|_{op}, \|D_2\|_{op}\}$$

$D_1$ and $D_2$ are defined below. Applying Lemma B.1 with dimension $\min\{d, M\}$, when $R_{\min} = \Omega(\sqrt{\min\{d, M\}})$,

$$\|D_1\|_{op} = \|\mathbb{E}w_1(\tilde{X};\tilde{\boldsymbol{\mu}}^u)(1-w_1(\tilde{X};\tilde{\boldsymbol{\mu}}^u))(\tilde{X}^{\min\{d,M\}}-\tilde{\boldsymbol{\mu}_1^u})(\tilde{X}^{\min\{d,M\}}-\tilde{\boldsymbol{\mu}_1^u})^T\|_{op}$$

$$\leq 2M\left(\frac{3}{2}R_{\max}+\min\{d,M\}\right)^2 \exp\left(-\left(\frac{R_{\min}}{2}-a\right)^2\sqrt{\min\{d,M\}}/8\right)$$

For $D_2$, by independence and Lemma B.1, when $R_{\min} = \Omega(\sqrt{\min\{d, M\}})$,

$$\|D_2\|_{op} = \left\|\mathbb{E}w_1(\tilde{X};\tilde{\boldsymbol{\mu}}^u)(1-w_1(\tilde{X};\tilde{\boldsymbol{\mu}}^u))\left(I_{[d-M]_+} + \sum_i \pi_i(v_i^{[d-M]_+})(v_i^{[d-M]_+})^T\right)\right\|_{op}$$

$$= \left\|\left(\mathbb{E}_{\tilde{X}_{\min\{d,M\}}}w_1(\tilde{X}_{\min\{d,M\}};\tilde{\boldsymbol{\mu}}^u)(1-w_1(\tilde{X}_{\min\{d,M\}};\tilde{\boldsymbol{\mu}}^u))\right)\right.$$

$$\left.\cdot\mathbb{E}_{X_{[d-M]_+}}\left(I_{[d-M]_+} + \sum_i \pi_i(v_i^{[d-M]_+})(v_i^{[d-M]_+})^T\right)\right\|_{op}$$

$$\leq (R_{\max}^2+1)2M\exp\left(-\left(\frac{R_{\min}}{2}-a\right)^2\sqrt{\min\{d,M\}}/8\right)$$

Combining the two and plugging in Eq. (B.17),

$$U_1 \leq 2M\exp\left(-\left(\frac{R_{\min}}{2}-a\right)^2\sqrt{\min\{d,M\}}/8\right)\cdot$$

$$\left(\max\left\{\left(\frac{3}{2}R_{\max}+\min\{d,M\}\right)^2, (R_{\max}^2+1)\right\} + a\left(\frac{3}{2}R_{\max}+\min\{d,M\}\right)\right)$$

$$\leq 2M(2R_{\max}+\min\{d,M\})^2\exp\left(-\left(\frac{R_{\min}}{2}-a\right)^2\sqrt{\min\{d,M\}}/8\right)$$

The max will always be achieved at the first term as $\min\{d, M\} \geq 1$. Similarly, with the same rotation, for $U_i, i \neq 1$,

$$U_i \leq \sup_u \|\mathbb{E}w_1(X;\boldsymbol{\mu}^u)w_i(X;\boldsymbol{\mu}^u)(X-\boldsymbol{\mu}_1^u)(X-\boldsymbol{\mu}_i^u)^T\|_{op} + a\|\mathbb{E}w_1(X;\boldsymbol{\mu}^u)w_i(X;\boldsymbol{\mu}^u)(X-\boldsymbol{\mu}_i^u)\|$$

By Lemma B.2, when $R_{\min} = \Omega(\sqrt{\min\{d, M\}})$, we have

$$U_i \leq \exp\left(-\left(\frac{R_{\min}}{2}-a\right)^2\sqrt{\min\{d,M\}}/8\right)\cdot$$

$$\left(\max\left\{(1+2\kappa)\left(\frac{3}{2}R_{\max}+\min\{d,M\}\right)^2, 2M(R_{\max}^2+1)\right\} + 2Ma\left(\frac{3}{2}R_{\max}+\min\{d,M\}\right)\right)$$

$$\leq \exp\left(-\left(\frac{R_{\min}}{2}-a\right)^2\sqrt{\min\{d,M\}}/8\right)\left(\frac{3}{2}R_{\max}+\min\{d,M\}\right)$$

$$\cdot\left(\max\{(1+2\kappa), 2M\}\left(\frac{3}{2}R_{\max}+\min\{d,M\}\right) + 2Ma\right)$$

$$\leq \exp\left(-\left(\frac{R_{\min}}{2}-a\right)^2\sqrt{\min\{d,M\}}/8\right)\left(\frac{3}{2}R_{\max}+\min\{d,M\}\right)^2\cdot\max\{3M, M+2\kappa+1\}$$

$$\leq M(2\kappa+4)\left(\frac{3}{2}R_{\max}+\min\{d,M\}\right)^2\exp\left(-\left(\frac{R_{\min}}{2}-a\right)^2\sqrt{\min\{d,M\}}/8\right)$$

The second inequality is because $R_{\max}^2 + 1 \le \left(\frac{3}{2}R_{\max} + \min\{d, M\}\right)^2$ and the third inequality is because $2a \le \frac{3}{2}R_{\max} + \min\{d, M\}$. Taking back to Eq. (B.16), and summing over $i \in [M]$, we have

$$\|\nabla_{\boldsymbol{\mu}_i} Q(\boldsymbol{\mu}|\boldsymbol{\mu}^t) - \nabla_{\boldsymbol{\mu}_i} q(\boldsymbol{\mu})\|$$

$$\le M(2\kappa + 4)\left(2R_{\max} + \min\{d, M\}\right)^2 \exp\left(-\left(\frac{R_{\min}}{2} - a\right)^2 \sqrt{\min\{d, M\}}/8\right) \sum_{i=1}^{M} \|\boldsymbol{\mu}_i^t - \boldsymbol{\mu}_i^*\|$$

This completes the proof. $\qquad\square$

## B.2 Proof of Theorem 1

*Proof of Theorem 1.* By Theorem 4 and Theorem 3, it suffices to check $\gamma \le \pi_{\min}$. Solving the inequality we have

$$a \le \frac{R_{\min}}{2} - \frac{2\sqrt{2}}{\sqrt[4]{\min\{d, M\}}}\sqrt{\log\left(\frac{M^2(2\kappa + 4)(2R_{\max} + \min\{d, M\})^2}{\pi_{\min}}\right)}$$

Combined with the condition in Theorem 4, we have

$$a \le \frac{R_{\min}}{2} - \max\left\{\frac{2\sqrt{2}}{\sqrt[4]{\min\{d, M\}}}\sqrt{\log\left(\frac{M^2(2\kappa + 4)(2R_{\max} + \min\{d, M\})^2}{\pi_{\min}}\right)},\right.$$

$$\left.\sqrt{\min\{d, M\}}\max(4\sqrt{2[\log(R_{\min}/4)]_+}, 8\sqrt{3})\right\}$$

$$= \frac{R_{\min}}{2} - \sqrt{\min\{d, M\}}o(R_{\min})$$

because

$$\max\left\{c\sqrt{\log(c_1\frac{M^2\kappa}{\pi_{\min}} + 2\log\left(2R_{\max} + \min\{d, M\}\right)}, \sqrt{\min\{d, M\}}\max\{c_2\sqrt{\log(R_{\min}/4)_+}, 8\sqrt{3}\}\right\}$$

$$\le \max\left\{c\sqrt{\log(c_1\frac{M^2\kappa}{\pi_{\min}} + c_2 R_{\max} + c_3\min\{d, M\})}, c'\sqrt{\min\{d, M\}}\sqrt{\log(R_{\max} + e)}\right\}$$

$$\le \sqrt{\min\{d, M\}}O\left(\sqrt{\log\left(\max\left\{\frac{M^2\kappa}{\pi_{\min}}, R_{\max}, \min\{d, M\}\right\}\right)}\right)$$

The condition in Theorem 4 can be rewritten as

$$a \le \frac{R_{\min}}{2} - \sqrt{\min\{d, M\}}O\left(\sqrt{\log\left(\max\left\{\frac{M^2\kappa}{\pi_{\min}}, R_{\max}, \min\{d, M\}\right\}\right)}\right)$$

$\qquad\square$

## C  Proofs for sample-based gradient EM

In this section we develop the error bound for sample-based gradient EM. Our proof is based on the Rademacher complexity theory and some new tools for contraction result. In [5], Maurer has the following contraction result for the complexity defined over countable sets.

**Lemma C.1** (Theorem 3 [5]). *Let $X$ be nontrivial, symmetric and sub-gaussian. Then there exists a constant $C < \infty$, depending only on the distribution of $X$, such that for any countable set $\mathcal{S}$ and function $h_i : \mathcal{S} \to \mathbb{R}$, $f_i : \mathcal{S} \to \mathbb{R}^k$, $i \in [n]$ satisfying $\forall s, s' \in \mathcal{S}, |h_i(s) - h_i(s')| \le L\|f(s) - f(s')\|$. If $\epsilon_{ik}$ is an independent doubly indexed Rademacher sequence, we have,*

$$\mathbb{E}\sup_{s \in \mathcal{S}}\sum_i \epsilon_i h_i(s) \le \mathbb{E}\sqrt{2}L\sup_{s \in \mathcal{S}}\sum_{i,k}\epsilon_{ik}f_i(s)_k,$$

*where $f_i(s)_k$ is the k-th component of $f_i(s)$.*

We prove Lemma 3 by generalizing this result to any subset of separable Banach space.

*Proof of Lemma 3.* First note that a subset of a separable subspace is separable, and has a dense countable subset; lets call this $\mathcal{S}_0$. Now note that if the Lipschitz condition holds for $s, s' \in \mathcal{S}$, then it also holds for $s, s' \in \mathcal{S}_0$. Now applying Lemma C.1, we see that

$$\mathbb{E} \sup_{s \in \mathcal{S}_0} \sum_i \epsilon_i h_i(s) \leq \mathbb{E}\sqrt{2} L \sup_{s \in \mathcal{S}_0} \sum_{i,k} \epsilon_{ik} f_i(s)_k,$$

All we need to prove is that the two supremas over $\mathcal{S}_0$ on the LHS and RHS of the above equation can be replaced by supremum over $\mathcal{S}$. We will only show this for the LHS. The argument for the RHS is identical. In order to show this, we need to also make sure that $g(s) := \sum_i \epsilon_i h_i(s)$ over $\mathcal{S}$ is measurable. We show this using standard tools from measure theory.

We want to show that:

$$\sup_{s \in \mathcal{S}} g(s) = \sup_{s \in \mathcal{S}_0} g(s). \tag{C.1}$$

Since $g(s)$ is continuous, its also measurable for all $s \in \mathcal{S}$. The above statement, once proven, essentially implies that the sup over $\mathcal{S}$ is the same as the sup over a countable set $\mathcal{S}_0$. Since pointwise sup over measurable functions is measurable, we are done. We now prove Eq. (C.1). It is clear that, $\sup_{s \in \mathcal{S}} g(s) \geq \sup_{s \in \mathcal{S}_0} g(s)$. So all we need is to prove that for all $\epsilon > 0$.

$$\sup_{s \in \mathcal{S}} g(s) \leq \sup_{s \in \mathcal{S}_0} g(s) + \epsilon \tag{C.2}$$

Since $g(s)$ is continuous, let $D_1(s) = \{s' \in \mathcal{S} : |g(s) - g(s')| \leq \epsilon\}$. Furthermore, since $\mathcal{S}_0$ is dense in $\mathcal{S}$, we also have $D_2(s, \epsilon) := D_1(s) \cap \mathcal{S}_0 \neq \phi$. So for each $s \in \mathcal{S}$, and $\epsilon > 0$, $\exists s' \in \mathcal{S}_0$ (to be precise, $s' \in D_2(s, \epsilon)$) such that $g(s) \leq g(s') + \epsilon$. Taking a sup over the LHS over $\mathcal{S}$ and a sup of RHS over $\mathcal{S}_0$, we get Eq. (C.2). This completes the proof. $\square$

*Proof of Proposition 1.* For any unit vector $u$, the Rademacher complexity of $\mathcal{F}$ is

$$R_n(\mathcal{F}) = \mathbb{E}_X \mathbb{E}_\epsilon \sup_{\boldsymbol{\mu} \in \mathbb{A}} \frac{1}{n} \sum_{i=1}^n \epsilon_i w_1(X_i; \boldsymbol{\mu}) \langle X_i - \boldsymbol{\mu}_1, u \rangle$$

$$\leq \underbrace{\mathbb{E}_X \mathbb{E}_\epsilon \sup_{\boldsymbol{\mu} \in \mathbb{A}} \frac{1}{n} \sum_{i=1}^n \epsilon_i w_1(X_i; \boldsymbol{\mu}) \langle X_i, u \rangle}_{(D)} + \underbrace{\mathbb{E}_X \mathbb{E}_\epsilon \sup_{\boldsymbol{\mu} \in \mathbb{A}} \frac{1}{n} \sum_{i=1}^n \epsilon_i w_1(X_i; \boldsymbol{\mu}) \langle \boldsymbol{\mu}_1, u \rangle}_{(E)} \tag{C.3}$$

We bound the two terms separately. Define $\eta_j(\boldsymbol{\mu}) : \mathbb{R}^{Md} \to \mathbb{R}^M$ to be a vector valued function with the $k$-th coordinate

$$[\eta_j(\boldsymbol{\mu})]_k = \frac{\|\boldsymbol{\mu}_1\|^2}{2} - \frac{\|\boldsymbol{\mu}_k\|^2}{2} + \langle X_j, \boldsymbol{\mu}_k - \boldsymbol{\mu}_1 \rangle + \log\left(\frac{\pi_k}{\pi_1}\right)$$

We claim

$$|w_1(X_j; \boldsymbol{\mu}) - w_1(X_j; \boldsymbol{\mu}')| \leq \frac{\sqrt{M}}{4} \|\eta_j(\boldsymbol{\mu}) - \eta_j(\boldsymbol{\mu}')\| \tag{C.4}$$

This vectorized Lipschitz condition simply follows from the fact that

$$w_1(X_j, \boldsymbol{\mu}) = \frac{1}{1 + \sum_{k=2}^M \exp([\eta_j(\boldsymbol{\mu})]_k)}$$

$$\frac{\partial w_1(X_j, \boldsymbol{\mu})}{\partial [\eta_j(\boldsymbol{\mu})]_k} = \frac{\exp([\eta_j(\boldsymbol{\mu})]_k)}{(1 + \sum_{k=2}^M \exp([\eta_j(\boldsymbol{\mu})]_k))^2} \leq \frac{1}{4}$$

so $w_1(X_j, \boldsymbol{\mu})$ is $\frac{1}{4}$-Lipschitz continuous w.r.t. $[\eta_j(\boldsymbol{\mu})]_k$. By Lemma A.8, $w_1(X_j, \boldsymbol{\mu})$ is $\frac{\sqrt{M}}{4}$ Lipschitz w.r.t $\eta_j(\boldsymbol{\mu})$. Now let $\psi_j(\boldsymbol{\mu}) = w_1(X_j; \boldsymbol{\mu}) \langle X_j, u \rangle$.

With Lipschitz property (C.4) and by Lemma C.1, we have

$$\mathbb{E}\left[\sup_{\boldsymbol{\mu}\in\mathbb{A}}\frac{1}{n}\sum_{j=1}^{n}\epsilon_j w_1(X_j;\boldsymbol{\mu})\langle X_j,u\rangle\right] \leq \mathbb{E}\left[\frac{1}{n}\sup_{\boldsymbol{\mu}\in\mathbb{A}}\sum_{j=1}^{n}\sum_{k=1}^{M}\epsilon_{jk}[\eta_j(\boldsymbol{\mu})]_k\frac{\sqrt{2M}}{4}\langle X_j,u\rangle\right]$$

$$=\mathbb{E}\left[\frac{\sqrt{2}M^{\frac{1}{2}}}{4n}\sup_{\boldsymbol{\mu}\in\mathbb{A}}\sum_{j=1}^{n}\sum_{k=2}^{M}\epsilon_{jk}\left(\frac{\|\boldsymbol{\mu}_1\|^2}{2}-\frac{\|\boldsymbol{\mu}_k\|^2}{2}+\langle X_j,\boldsymbol{\mu}_k-\boldsymbol{\mu}_1\rangle+\log(\frac{\pi_k}{\pi_1})\right)\langle X_j,u\rangle\right]$$

$$\leq\mathbb{E}\left[\underbrace{\frac{\sqrt{2M}}{4n}\sup_{\boldsymbol{\mu}\in\mathbb{A}}\sum_{j=1}^{n}\sum_{k=1}^{M}\epsilon_{jk}\left(\frac{\|\boldsymbol{\mu}_1\|^2}{2}-\frac{\|\boldsymbol{\mu}_k\|^2}{2}+\log(\frac{\pi_k}{\pi_1})\right)\langle X_j,u\rangle}_{(D.1)}\right]\quad\text{(C.5)}$$

$$+\mathbb{E}\left[\underbrace{\frac{\sqrt{2M}}{4n}\sup_{\boldsymbol{\mu}\in\mathbb{A}}\sum_{j=1}^{n}\sum_{k=1}^{M}\epsilon_{jk}\langle X_j,\boldsymbol{\mu}_k-\boldsymbol{\mu}_1\rangle\langle X_j,u\rangle}_{(D.2)}\right]$$

To bound $(D.1)$, note that the sum over $k=1,\cdots,M$ can be considered as an inner product of two vectors in $\mathbb{R}^M$. The supremum of $\|\boldsymbol{\mu}\|$ can be bounded as $\max_{\boldsymbol{\mu}\in\mathbb{A}}\|\boldsymbol{\mu}_i\|\leq\|\boldsymbol{\mu}_{\max}^*\|+a\leq\frac{3}{2}R_{\max}$.

$$(D.1)=\mathbb{E}\left[\frac{\sqrt{2M}}{4}\sup_{\boldsymbol{\mu}\in\mathbb{A}}\begin{pmatrix}\frac{\|\boldsymbol{\mu}_1\|^2}{2}-\frac{\|\boldsymbol{\mu}_1\|^2}{2}+\log(\frac{\pi_1}{\pi_1})\\\vdots\\\frac{\|\boldsymbol{\mu}_1\|^2}{2}-\frac{\|\boldsymbol{\mu}_M\|^2}{2}+\log(\frac{\pi_M}{\pi_1})\end{pmatrix}^T\begin{pmatrix}\frac{1}{n}\sum_{j=1}^{n}\epsilon_{j1}\langle X_j,u\rangle\\\vdots\\\frac{1}{n}\sum_{j=1}^{n}\epsilon_{jM}\langle X_j,u\rangle\end{pmatrix}\right]$$

$$\leq cM(9R_{\max}^2/4+\log(\kappa))\mathbb{E}\left[\left\|\begin{pmatrix}\frac{1}{n}\sum_{j=1}^{n}\epsilon_{j1}\langle X_j,u\rangle\\\vdots\\\frac{1}{n}\sum_{j=1}^{n}\epsilon_{jM}\langle X_j,u\rangle\end{pmatrix}\right\|\right]\quad\text{(C.6)}$$

By Lemma A.6, and $\|u\|=1$, we know $\langle X_j,u\rangle$ is sub-Gaussian with parameter upper bounded by $1+R_{\max}$. So each element of the vector in Equation C.6 is the average of $n$ independent mean 0 sub-Gaussian random variables with sub-gaussian norm upper bounded by $1+R_{\max}$ (since w.l.o.g we have assumed that $\sigma=1$ and $\max_i\|\mu\|\leq R_{\max}$, by Lemma A.7). Consequently, $\forall k\in[M]$, $\mathbb{E}\left|\frac{1}{n}\sum_{j=1}^{n}\epsilon_{jk}\langle X_j,u_1\rangle\right|\leq c(1+R_{\max})/\sqrt{n}$ for some global constant $c$ [7], and

$$(D.1)\leq cM^{3/2}(9R_{\max}^2/4+\log(\kappa))(1+R_{\max})\frac{1}{\sqrt{n}}\leq cM^{3/2}(1+R_{\max})^3\max\{1,\log(\kappa)\}\frac{1}{\sqrt{n}}$$

On the other hand, for $(D.2)$, we have

$$(D.2)=\mathbb{E}\left[\frac{\sqrt{2M}}{4n}\sup_{\boldsymbol{\mu}\in\mathbb{A}}\sum_{j=1}^{n}\sum_{k=1}^{M}\epsilon_{jk}\langle X_j,\boldsymbol{\mu}_k-\boldsymbol{\mu}_1\rangle\langle X_j,u\rangle\right]$$

$$=\mathbb{E}\left[\frac{\sqrt{2M}}{4n}\sup_{\boldsymbol{\mu}\in\mathbb{A}}\sum_{k=1}^{M}(\boldsymbol{\mu}_k-\boldsymbol{\mu}_1)^T\left(\sum_{j=1}^{n}\epsilon_{jk}X_jX_j^T\right)u\right]$$

$$\leq\sum_{k=1}^{M}\mathbb{E}\left[\frac{\sqrt{2M}}{4}\sup_{\boldsymbol{\mu}\in\mathbb{A}}\|\boldsymbol{\mu}_k-\boldsymbol{\mu}_1\|\left\|\frac{1}{n}\sum_{j=1}^{n}\epsilon_{jk}X_jX_j^T\right\|_{op}\right]\quad\text{(C.7)}$$

$$\leq\sum_{k=1}^{M}\frac{\sqrt{2M}}{2}\|\boldsymbol{\mu}_{\max}\|\mathbb{E}\left[\left\|\frac{1}{n}\sum_{j=1}^{n}\epsilon_{jk}X_jX_j^T\right\|_{op}\right]$$

For each $k\in[M]$, the operator norm $\|\frac{1}{n}\sum_{j=1}^{n}\epsilon_{jk}X_jX_j^T\|_{op}$ can be bounded by the same discretization technique with the $1/2$-covering of the unit sphere. To be specific, since for any matrix

$A$, $\|A\|_{op} = \sup_{u \in \mathcal{S}^{d-1}} \|Au\|$,

$$\forall u, \exists u_j \text{ s.t. } \|Au\| \leq \|Au_j\| + \|A\|_{op}\|u - u_j\| \leq \max_j \|Au_j\| + \frac{1}{2}\|A\|_{op}$$

Taking $\sup_{u \in \mathcal{S}^{d-1}}$ on the left side, we get $\|A\|_{op} \leq 2\max_j \|Au_j\|$. Therefore $\|\frac{1}{n}\sum_{j=1}^{n} \epsilon_{jk} X_j X_j^T\|_{op} \leq 2\max_\ell \frac{1}{n}\sum_{j=1}^{n} \epsilon_{jk}\langle X_j, u_\ell\rangle^2$. The square of sub-gaussian random variable $\langle X_j, u_\ell\rangle$ is sub-exponential, from Lemma 5.14 in [7] we know

$$\mathbb{E}\left[\exp\left(\frac{1}{n}\sum_{j=1}^{n}\epsilon_{jk}\langle X_j, u\rangle^2 t\right)\right] \leq \exp\left(\frac{c_4 t^2 (1 + R_{\max})^4}{n}\right)$$

With the 1/2-covering number of $\mathcal{S}^{d-1}$ bounded by $\exp(2d)$, we have

$$\mathbb{E}\left[\exp\left(t \cdot \|\frac{1}{n}\sum_{j=1}^{n}\epsilon_{jk}X_j X_j^T\|_{op}\right)\right] \leq \exp\left(2d + \frac{c_5 t^2 (1 + R_{\max})^4}{n}\right)$$

Hence,

$$\mathbb{E}\left[\left\|\frac{1}{n}\sum_{j=1}^{n}\epsilon_{jk}X_j X_j^T\right\|_{op}\right] = \frac{1}{t}\log\left(\exp\left(t\mathbb{E}\left[\left\|\frac{1}{n}\sum_{j=1}^{n}\epsilon_{jk}X_j X_j^T\right\|_{op}\right]\right)\right), \quad \forall t > 0$$

$$\leq \frac{1}{t}\log\left(\mathbb{E}\left[\exp\left(t\left\|\frac{1}{n}\sum_{j=1}^{n}\epsilon_{jk}X_j X_j^T\right\|_{op}\right)\right]\right)$$

$$\leq \frac{2d}{t} + \frac{ct(1 + R_{\max})^4}{n}$$

Taking $t = c\frac{\sqrt{nd}}{(1+R_{\max})^2}$,

$$\mathbb{E}\left[\left\|\frac{1}{n}\sum_{j=1}^{n}\epsilon_{jk}X_j X_j^T\right\|_{op}\right] \leq c\sqrt{\frac{d}{n}}(1 + R_{\max})^2$$

Plugging back to Eq. (C.7), and use $\sup_{\boldsymbol{\mu} \in \mathbb{A}}\|\boldsymbol{\mu}\| \leq \sup_k \|\boldsymbol{\mu}_k^*\| + a \leq \frac{3}{2}R_{\max}$, we have

$$(D.2) \leq \frac{cM(1 + R_{\max})^3 \sqrt{d}}{\sqrt{n}}$$

Plugging the bound back to Eq. (C.5), we have

$$(D) \leq \frac{cM^{3/2}(1 + R_{\max})^3 \sqrt{d}\max\{1, \log(\kappa)\}}{\sqrt{n}}$$

Apply Lemma C.1 on the $(E)$ term in Eq. (C.3), we have

$$(E) = \mathbb{E}\left[\sup_{\boldsymbol{\mu} \in \mathbb{A}} \frac{1}{n}\sum_{j=1}^{n}\epsilon_j w_i(X_j; \boldsymbol{\mu})\langle \boldsymbol{\mu}_i, u\rangle\right]$$

$$\leq \mathbb{E}\left[\frac{\sqrt{2M}}{4n}\sup_{\boldsymbol{\mu} \in \mathbb{A}}\sum_{j=1}^{n}\sum_{k=1}^{M}\epsilon_{jk}\left(\frac{\|\boldsymbol{\mu}_1\|^2}{2} - \frac{\|\boldsymbol{\mu}_k\|^2}{2} + \langle X_j, \boldsymbol{\mu}_k - \boldsymbol{\mu}_1\rangle + \log(\frac{\pi_k}{\pi_1})\right)\langle \boldsymbol{\mu}_i, u\rangle\right]$$

$$\leq \underbrace{\frac{\sqrt{2M}}{4}\mathbb{E}_\epsilon\left[\sup_{\boldsymbol{\mu} \in \mathbb{A}} \frac{1}{n}\sum_{j=1}^{n}\sum_{k=1}^{M}\epsilon_{jk}\left(\frac{\|\boldsymbol{\mu}_1\|^2}{2} - \frac{\|\boldsymbol{\mu}_k\|^2}{2} + \log\frac{\pi_k}{\pi_1}\right)\langle \boldsymbol{\mu}_i, u\rangle\right]}_{E.1}$$

$$+ \underbrace{\frac{\sqrt{2M}}{4}\mathbb{E}_{X,\epsilon}\left[\sup_{\boldsymbol{\mu} \in \mathbb{A}} \frac{1}{n}\sum_{j=1}^{n}\sum_{k=1}^{M}\epsilon_{jk}\langle X_j, \boldsymbol{\mu}_k - \boldsymbol{\mu}_1\rangle\langle \boldsymbol{\mu}_i, u\rangle\right]}_{E.2}$$

We will now bound $(E.1)$ and $(E.2)$.

$$(E.1) \leq \frac{\sqrt{2M}}{4} \mathbb{E}_{\epsilon} \left[ \sup_{\boldsymbol{\mu} \in \mathbb{A}} \frac{1}{n} \sum_{j=1}^{n} \sum_{k=1}^{M} \epsilon_{jk} \left( \frac{\|\boldsymbol{\mu}_1\|^2}{2} - \frac{\|\boldsymbol{\mu}_k\|^2}{2} + \log \frac{\pi_k}{\pi_1} \right) \sup_{\boldsymbol{\mu} \in \mathbb{A}} \langle \boldsymbol{\mu}_i, u \rangle \right]$$

$$\leq \frac{\sqrt{2M}}{4} R_{\max} \mathbb{E}_{\epsilon} \left[ \sup_{\boldsymbol{\mu} \in \mathbb{A}} \begin{pmatrix} \frac{\|\boldsymbol{\mu}_1\|^2}{2} - \frac{\|\boldsymbol{\mu}_1\|^2}{2} + \log(\frac{\pi_1}{\pi_1}) \\ \vdots \\ \frac{\|\boldsymbol{\mu}_1\|^2}{2} - \frac{\|\boldsymbol{\mu}_M\|^2}{2} + \log(\frac{\pi_M}{\pi_1}) \end{pmatrix}^T \begin{pmatrix} \frac{1}{n} \sum_{j=1}^{n} \epsilon_{j1} \\ \vdots \\ \frac{1}{n} \sum_{j=1}^{n} \epsilon_{jM} \end{pmatrix} \right]$$

$$\leq c M R_{\max} (9 R_{\max}^2 / 4 + \log \kappa) \mathbb{E}_{\epsilon} \left\| \begin{pmatrix} \frac{1}{n} \sum_{j=1}^{n} \epsilon_{j1} \\ \vdots \\ \frac{1}{n} \sum_{j=1}^{n} \epsilon_{jM} \end{pmatrix} \right\| \tag{C.8}$$

Note that each element of the vector in Equation C.8 is the average of $n$ i.i.d mean 0 Radamacher random variables, which are essentially sub-gaussian radnom variables with subgaussian norm upper bounded by 1. Consequently, $\forall k \in [M]$, $\mathbb{E} \left| \frac{1}{n} \sum_{j=1}^{n} \epsilon_{jk} \right| \leq c'/\sqrt{n}$ for some global constant $c$ [7], and

$$(E.1) \leq c' M^{3/2} R_{\max} (9 R_{\max}^2 / 4 + \log \kappa)/\sqrt{n}$$

As for (E.2), we have

$$(E.2) \leq \frac{\sqrt{2M}}{4} \mathbb{E}_{X,\epsilon} \left[ \sup_{\boldsymbol{\mu} \in \mathbb{A}} \frac{1}{n} \sum_{j=1}^{n} \sum_{k=1}^{M} \epsilon_{jk} \langle X_j, \boldsymbol{\mu}_k - \boldsymbol{\mu}_1 \rangle \sup_{\boldsymbol{\mu} \in \mathbb{A}} \langle \boldsymbol{\mu}_i, u \rangle \right]$$

$$\leq \frac{3\sqrt{2M}}{8} R_{\max} \mathbb{E}_{X,\epsilon} \left[ \sup_{\boldsymbol{\mu} \in \mathbb{A}} \sum_{k=1}^{M} (\boldsymbol{\mu}_k - \boldsymbol{\mu}_1)^T \left( \frac{1}{n} \sum_{j=1}^{n} \epsilon_{jk} X_j \right) \right]$$

$$\leq \frac{3\sqrt{2M}}{8} R_{\max} \sum_{k=1}^{M} \mathbb{E}_{X,\epsilon} \left[ \sup_{\boldsymbol{\mu} \in \mathbb{A}} (\boldsymbol{\mu}_k - \boldsymbol{\mu}_1)^T \left( \frac{1}{n} \sum_{j=1}^{n} \epsilon_{jk} X_j \right) \right]$$

$$\leq \frac{9\sqrt{2M}}{8} R_{\max}^2 \sum_{k=1}^{M} \mathbb{E}_{X,\epsilon} \left\| \frac{1}{n} \sum_{j=1}^{n} \epsilon_{jk} X_j \right\| \tag{C.9}$$

In Eq (C.9), the vector $\frac{1}{n} \sum_{j=1}^{n} \epsilon_{jk} X_j$ is the average of $n$ independent mean zero isotropic subgaussian random vectors. Another using of the discretizing technique along with the moment generating function with $t \geq 0$ gives:

$$\left\| \frac{1}{n} \sum_{j=1}^{n} \epsilon_{jk} X_j \right\| \leq 2 \max_{\ell} \langle \frac{1}{n} \sum_{j=1}^{n} \epsilon_{jk} X_j, u_\ell \rangle$$

$$E \left[ \exp t \left\| \frac{1}{n} \sum_{j=1}^{n} \epsilon_{jk} X_j \right\| \right] \leq \sum_{\ell} E \left[ \exp \left( 2\frac{t}{n} \sum_{j=1}^{n} \epsilon_{jk} \langle X_j, u_\ell \rangle \right) \right] \leq \exp \left( 2d + \frac{c'(1+R_{\max})^2 t^2}{n} \right)$$

$$E \left\| \frac{1}{n} \sum_{j=1}^{n} \epsilon_{jk} X_j \right\| \leq \frac{c'' + 2d + \frac{c'(1+R_{\max})^2 t^2}{n}}{t} \qquad \text{Using Jensen's inequality}$$

Taking $t = \Theta\sqrt{nd}/(1 + R_{\max})$,

$$(E.2) \leq c M^{3/2} R_{\max}^2 (1 + R_{\max})\sqrt{d}/\sqrt{n}$$

Thus, combing (E.1) and (E.2) we get:

$$(E) \leq \frac{c M^{3/2} (1 + R_{\max})^3 \max\{1, \log(\kappa)\}\sqrt{d}}{\sqrt{n}}$$

The final bound follows by combining (D) and (E):

$$R_n(\mathcal{F}) \leq \frac{cM^{3/2}(1 + R_{\max})^3 \sqrt{d}\max\{1, \log(\kappa)\}}{\sqrt{n}}$$

<div style="text-align:right">□</div>

For proving Lemma 2 we first recall the following symmetrization lemma in learning theory.

**Lemma C.2** (See e.g. [6])**.** *Let $\mathcal{F}$ be a function class with domain X. Let $\{X_1, X_2, \cdots, X_n\}$ be a set of sample generated by a distribution $\mathbb{P}$ on X. Assume $\sigma_i$ are i.i.d. Rademacher variables, then*

$$\mathbb{E}\left(\sup_{f \in \mathcal{F}}(\mathbb{E}f - \frac{1}{n}\sum_{i=1}^{n} f(X_i))\right) \leq 2R_n(\mathcal{F})$$

*Here $R_n(\mathcal{F}) = \mathbb{E}\left[\sup_{f \in \mathcal{F}}|\frac{1}{n}\sum_{i=1}^{n}\sigma_i f(X_i)\right]$ is the Rademacher complexity.*

*Proof of Lemma 2.* Consider for some $r > 0$, the set in which $X$ lies in the $r$-ball of its corresponding center. If $Z_i$ denotes the hidden cluster assignment of $X_i$, we denote $\mathcal{Y}_r^i = \{X_1, \cdots, X_n : \|X_i - \boldsymbol{\mu}_{Z_i}^*\| \leq r\}$.

$$\mathcal{Y}_r := \{X_1, \dots X_n : \|X_i - \boldsymbol{\mu}_{Z_i}^*\| \leq r, \ , \forall i \in [n]\} = \cap_i \mathcal{Y}_r^i$$

By Lemma A.5 and union bound, for $r = \Omega(\sqrt{d})$,

$$p := P(\mathbf{X} \notin \mathcal{Y}_r) \leq \sum_{i=1}^{n} P(X \in (\mathcal{Y}_r^i)^c) \leq cn \exp\left(-\frac{r\sqrt{d}}{2}\right). \tag{C.10}$$

Let $m_r := \mathbb{E}[g(X)|X \in \mathcal{Y}_r]$, we want to show $m_r$ is close to $\mathbb{E}[g(\mathbf{X})]$ and is close to $g(\mathbf{X})$ with high probability.

Let $\mathbf{X}$ and $\mathbf{X}'$ be two samples which only differ on one data-point, then

$$g(\mathbf{X}) - g(\mathbf{X}') = \sup_{\boldsymbol{\mu} \in \mathbb{A}}\left(\frac{1}{n}\sum_{i=1}^{n} w_1(X_i; \boldsymbol{\mu})\langle X_i - \boldsymbol{\mu}_1, u\rangle - \mathbb{E}_X w_1(X; \boldsymbol{\mu})\langle X - \boldsymbol{\mu}_1, u\rangle\right)$$

$$- \sup_{\boldsymbol{\mu} \in \mathbb{A}}\left(\frac{1}{n}\sum_{i=1}^{n} w_1(X_i'; \boldsymbol{\mu})\langle X_i' - \boldsymbol{\mu}_1, u\rangle - \mathbb{E}_X w_1(X'; \boldsymbol{\mu})\langle X' - \boldsymbol{\mu}_1, u\rangle\right)$$

Assume $\tilde{\boldsymbol{\mu}}$ be the maximizer for the supremum of $X$, then

$$g(\mathbf{X}) - g(\mathbf{X}') \overset{(i)}{\leq} \frac{1}{n}(\sum_{i=1}^{n} w_1(X_i; \tilde{\boldsymbol{\mu}})\langle X_i - \tilde{\boldsymbol{\mu}}_1, u\rangle - \mathbb{E}w_1(X; \tilde{\boldsymbol{\mu}})\langle X - \tilde{\boldsymbol{\mu}}_1, u\rangle)$$

$$- \frac{1}{n}\sum_{i=1}^{n}(w_1(X_i'; \tilde{\boldsymbol{\mu}})\langle X_i' - \tilde{\boldsymbol{\mu}}_1, u\rangle - \mathbb{E}w_1(X'; \tilde{\boldsymbol{\mu}})\langle X' - \tilde{\boldsymbol{\mu}}_1, u\rangle)$$

$$= \frac{1}{n}w_1(X_i; \tilde{\boldsymbol{\mu}})\langle X_i - \tilde{\boldsymbol{\mu}}_1, u\rangle - w_1(X_i'; \tilde{\boldsymbol{\mu}})\langle X_i' - \tilde{\boldsymbol{\mu}}_1, u\rangle$$

where (i) is by definition of supremum. The inequality holds when we change the order of $X$ and $X'$, hence for $X, X' \in \mathcal{Y}_r$,

$$|g(\mathbf{X}) - g(\mathbf{X}')| \leq \frac{1}{n}|w_1(X_i; \tilde{\boldsymbol{\mu}})\langle X_i - \tilde{\boldsymbol{\mu}}_1, u\rangle - w_1(X_i'; \tilde{\boldsymbol{\mu}})\langle X_i' - \tilde{\boldsymbol{\mu}}_1, u\rangle|$$

$$\leq \frac{2}{n}\sup_{\boldsymbol{\mu} \in \mathbb{A}, X \in \mathcal{Y}_r} |w_1(X_i; \boldsymbol{\mu})\langle X_i - \boldsymbol{\mu}_1, u\rangle|$$

$$\leq \frac{2}{n}\sup_{X \in \mathcal{Y}_r} (\|X - \boldsymbol{\mu}_{Z_X}^*\| + R_{\max})$$

$$\leq \frac{2(r + R_{\max})}{n} := L$$

By Theorem 6, we have

$$P(g(\mathbf{X}) - m_r \geq \epsilon) \leq p + \exp\left(-2\frac{(\epsilon - nLp)_+^2}{nL^2}\right)$$

$$\leq c_1 n \exp(-c\sqrt{d}r) + \exp\left(-2\frac{(\epsilon - nL \cdot c_1 n \exp(-c\sqrt{d}r))_+^2}{nL^2}\right)$$

$$= \underbrace{cn \exp(-c\sqrt{d}r)}_{P_1} + \underbrace{\exp\left(-\frac{c_1 n(\epsilon - c_2 n(r + R_{\max})\exp(-c\sqrt{d}r))_+^2}{(r + R_{\max})^2}\right)}_{P_2}$$

(C.11)

Let $r = \Theta((1 + R_{\max})\log^2(n)\sqrt{d})$ and $\epsilon = c_0(1 + R_{\max})d\log^{5/2}(n)/\sqrt{n}$. Since, for large $n$,

$$n(r + R_{\max})\exp(-cr\sqrt{d}) \leq c_2(1 + R_{\max})\exp(\log n + \log\log n - c\log^2 n) = o((1 + R_{\max})/\sqrt{n})$$

for some constant $c_0$, which yields for large $n$, $(\epsilon - c_2 n(r + R_{\max})\exp(-cr\sqrt{d}))_+ \geq \epsilon/2$. Finally, for large $n$, we can have the following bounds on $P_1$ and $P_2$.

$$P_1 = O\left(\exp(\log n - c(1 + R_{\max})^2(\log n)^2)\right) = O\left(\exp(-c'(1 + R_{\max})^2 d\log n)\right)$$

$$P_2 \leq \exp\left(-\frac{cn\epsilon^2}{d(\log^2 n(1 + R_{\max}))^2}\right) = O\left(\exp(-c''d\log n)\right)$$

(C.12)

where $c, c', c'''$ are some global constants. The last line uses the fact $r + R_{\max} = O(\sqrt{d}(1 + R_{\max})\log^2 n)$.

Now we bound the difference between $\mathbb{E}g(X)$ and the conditional expectation $m_r$. By the total expectation theorem,

$$\mathbb{E}g(\mathbf{X}) = m_r P(\mathbf{X} \in \mathcal{Y}_r) + \mathbb{E}[g(\mathbf{X})\mathbf{1}(\mathbf{X} \notin \mathcal{Y}_r)]$$

$$\mathbb{E}[g(\mathbf{X})](P(\mathbf{X} \in \mathcal{Y}_r) + P(\mathbf{X} \notin \mathcal{Y}_r)) = m_r P(\mathbf{X} \in \mathcal{Y}_r) + \mathbb{E}[g(\mathbf{X})\mathbf{1}(\mathbf{X} \notin \mathcal{Y}_r)]$$

$$\mathbb{E}g(\mathbf{X}) - m_r = \frac{\mathbb{E}[g(\mathbf{X})\mathbf{1}(\mathbf{X} \notin \mathcal{Y}_r)] - \mathbb{E}[g(\mathbf{X})]P(X \notin \mathcal{Y}_r)}{P(\mathbf{X} \in \mathcal{Y}_r)}$$

(C.13)

$$\Rightarrow |m_r - \mathbb{E}g(\mathbf{X})| \leq \frac{p|\mathbb{E}g(\mathbf{X})| + |\mathbb{E}[g(\mathbf{X})\mathbf{1}(\mathbf{X} \notin \mathcal{Y}_r)]|}{1 - p}$$

$p$ is defined in Eq (C.10). Note that by Proposition 1, and the symmetrization result Lemma C.2, $\mathbb{E}g(X) \leq 2R_n(\mathcal{F}) \leq cn^{-1/2}M^3\sqrt{d}(1 + R_{\max})^3\max\{1, \log(\kappa)\}$. On the other hand, as $g(\mathbf{X})$ is the sup over a class of quantity, which is centered at zero. So $g(X) \geq 0$. We also have $\mathbf{1}(\mathbf{X} \in \cup_i(\mathcal{Y}_r^i)^c) \leq \sum_{i=1}^n \mathbf{1}(\mathbf{X} \in (\mathcal{Y}_r^i)^c)$. Hence,

$$\mathbb{E}[g(\mathbf{X})\mathbf{1}(\mathbf{X} \notin \mathcal{Y}_r)] = \mathbb{E}[g(\mathbf{X})\sum_{i=1}^n \mathbf{1}(\mathbf{X} \in (\mathcal{Y}_r^i)^c)] \leq \sum_{i=1}^n \mathbb{E}[g(\mathbf{X})\mathbf{1}(\mathbf{X} \in (\mathcal{Y}_r^i)^c)]$$

Note for each sample $X_i$ and $\boldsymbol{\mu}$, $\left|\sup_{\boldsymbol{\mu}\in\mathbb{A}} w_1(X_i; \boldsymbol{\mu})\langle X_i - \boldsymbol{\mu}_1, u\rangle\right| \leq \sup_{\boldsymbol{\mu}\in\mathbb{A}} w_1(X_i; \boldsymbol{\mu})\|X_i - \boldsymbol{\mu}_{Z_i}^*\| + \|\boldsymbol{\mu}_{Z_i}^* - \boldsymbol{\mu}_1\| \leq \|X_i - \boldsymbol{\mu}_{Z_i}^*\| + 2R_{\max}$. Thus,

$$|g(\mathbf{X})| = |\sup_{\boldsymbol{\mu}\in\mathbb{A}} \frac{1}{n}\sum_{j=1}^n w_1(X_j; \boldsymbol{\mu})\langle X_j - \boldsymbol{\mu}_1, u\rangle - \mathbb{E}_X w_1(X; \boldsymbol{\mu})\langle X - \boldsymbol{\mu}_1, u\rangle|$$

$$\leq \frac{1}{n}\sum_{j=1}^n (\|X_j - \boldsymbol{\mu}_{Z_j}^*\| + 2R_{\max}) + \mathbb{E}_X\|X - \boldsymbol{\mu}_{Z_X}^*\| + 2R_{\max}$$

$$\leq \frac{1}{n}\sum_{j=1}^n \|X_j - \boldsymbol{\mu}_{Z_j}^*\| + \mathbb{E}_X\|X - \boldsymbol{\mu}_{Z_X}^*\| + 4R_{\max}$$

Therefore we have,

$$\mathbb{E}[g(\mathbf{X})1(\mathbf{X} \notin \mathcal{Y}_r)] \leq \sum_{i=1}^{n} \mathbb{E}_{\mathbf{X}}[(\frac{1}{n}\sum_{j=1}^{n}\|X_j - \boldsymbol{\mu}_{Z_j}^*\| + \mathbb{E}_X\|X - \boldsymbol{\mu}_{Z_X}^*\| + 4R_{\max})1(\mathbf{X} \in (\mathcal{Y}_r^i)^c)]$$

$$\leq \sum_{i=1}^{n} \mathbb{E}_{\mathbf{X}}[(\frac{1}{n}\sum_{j=1}^{n}\|X_j - \boldsymbol{\mu}_{Z_j}^*\|1(\mathbf{X} \in (\mathcal{Y}_r^i)^c)]] + (\mathbb{E}_X\|X - \boldsymbol{\mu}_{Z_X}^*\| + 4R_{\max})P(\mathbf{X} \in (\mathcal{Y}_r^i)^c)]$$

$$\leq \sum_{i=1}^{n} \frac{1}{n}\sum_{j=1}^{n}\mathbb{E}_{\mathbf{X}}[\|X_j - \boldsymbol{\mu}_{Z_j}^*\|1(\mathbf{X} \in (\mathcal{Y}_r^i)^c)] + c'(R_{\max} + d)p$$

(C.14)

where the last inequality follows from Lemma A.2. Note that when $j \neq i$, the expectation factors due to independence of the sample points and by Lemma A.5,

$$\mathbb{E}_{\mathbf{X}}[\|X_j - \boldsymbol{\mu}_{Z_j}^*\|1(\mathbf{X} \in (\mathcal{Y}_r^i)^c)] = \mathbb{E}_{X_j}\|X_j - \boldsymbol{\mu}_{Z_j}^*\| \cdot P(\|X_i - \boldsymbol{\mu}_{Z_i}^*\| \geq r) \leq cde^{-\frac{r\sqrt{d}}{2}}$$

When $j = i$, from Lemma A.4,

$$\mathbb{E}_{\mathbf{X}}[\|X_j - \boldsymbol{\mu}_{Z_j}^*\|1(\mathbf{X} \in (\mathcal{Y}_r^i)^c)] \leq cn\int_{v=r}^{\infty} v \cdot v^{d-1}(v + R_{\max} + a)\exp(-v^2/2)dv \cdot \frac{2\pi^{d/2}}{\Gamma\left(\frac{d}{2}\right)}$$

$$\leq c_1 d\exp\left(-\frac{r\sqrt{d}}{2}\right)$$

Putting back to Eq. (C.14), we have

$$\mathbb{E}[g(\mathbf{X})1(\mathbf{X} \notin \mathcal{Y}_r)] \leq c_1 nd\exp\left(-\frac{r\sqrt{d}}{2}\right) + c_2 d\exp\left(-\frac{r\sqrt{d}}{2}\right) + c_3 n(R_{\max} + d)\exp\left(-\frac{r\sqrt{d}}{2}\right)$$

$$\leq cn(R_{\max} + d)\exp\left(-\frac{r\sqrt{d}}{2}\right)$$

Following from Eq. (C.13), we have

$$|m_r - \mathbb{E}g(X)| \leq \frac{c_1 n\exp(-\frac{r}{2}\sqrt{d})R_n(\mathcal{F}) + c_2 n(R_{\max} + d)\exp(-\frac{r}{2}\sqrt{d})}{1 - c_3 n\exp(-c_4 r\sqrt{d})} \qquad \text{(C.15)}$$

Recall that we take $r = \Theta(\sqrt{d}(1 + R_{\max})\log^2 n)$, for large enough $n$, we have $1 - c_3 n\exp(-cr\sqrt{d}) \geq 1/2$, and $ne^{-cr\sqrt{d}} \leq C/n$. Finally for the second part of the numerator in Eq. (C.15) we have:

$$n(R_{\max} + d)\exp(-\sqrt{d}r/2) \leq (R_{\max} + 1)\exp(\log n + \log d - \Theta(d(1 + R_{\max})\log^2 n))$$

$$\leq C'(R_{\max} + 1)/\sqrt{n}.$$

Eq (C.15) becomes,

$$m_r \leq 2R_n(\mathcal{F})(1 + O(1/n)) + O((R_{\max} + 1)/\sqrt{n}) \qquad \text{(C.16)}$$

Thus using Eqs (C.11), (C.12) and (C.16) the final bound becomes:

$$P(g(X) \leq 2R_n(\mathcal{F})(1 + O(1/n)) + O((R_{\max} + 1)/\sqrt{n}) + (1 + R_{\max})d\sqrt{\log^5 n/n})$$

$$\geq 1 - P_1 - P_2$$

$$\geq 1 - c\exp\left(-c'\min((1 + R_{\max})^2 d\log n, d\log n)\right) \geq 1 - \exp\left(-cd\log n\right)$$

Finally we have,

$$P(g(\mathbf{X}) = \tilde{O}(\max\{R_n(\mathcal{F}), (1 + R_{\max})d\log^{5/2}(n)/\sqrt{n}\})) \geq 1 - \exp\left(-cd\log n\right)$$

$\square$

*Proof of Theorem 5.* Denote $Z_i = \sup_{\boldsymbol{\mu} \in \mathbb{A}} \left\| G^{(i)}(\boldsymbol{\mu}) - G_n^{(i)}(\boldsymbol{\mu}) \right\|$ where

$$G(\boldsymbol{\mu}) = \begin{pmatrix} \mathbb{E} w_1(X; \boldsymbol{\mu})(X - \boldsymbol{\mu}_1) \\ \mathbb{E} w_2(X; \boldsymbol{\mu})(X - \boldsymbol{\mu}_2) \\ \vdots \\ \mathbb{E} w_M(X; \boldsymbol{\mu})(X - \boldsymbol{\mu}_3) \end{pmatrix}.$$

Assume $\mathcal{S}^{d-1}$ is $d$-dimensional unit sphere. Recall the definition $g_i^u(X) = \sup_{\boldsymbol{\mu} \in \mathbb{A}} \langle G^{(i)}(\boldsymbol{\mu}) - G_n^{(i)}(\boldsymbol{\mu}), u \rangle = \sup_{\boldsymbol{\mu} \in \mathbb{A}} \frac{1}{n} \sum_{i=1}^n w_1(X_i; \boldsymbol{\mu}) \langle X_i - \boldsymbol{\mu}_1, u \rangle - \mathbb{E} w_1(X; \boldsymbol{\mu}) \langle X - \boldsymbol{\mu}_1, u \rangle$. Then $Z_i = \sup_{u \in \mathcal{S}^{d-1}} g_i^u(X)$. Without loss of generality, we assume $i = 1$, the proof for other clusters follows similarly. Let $\{u^{(1)}, u^{(2)}, \cdots, u^{(K)}\}$ be a $\frac{1}{2}$-covering of the unit sphere $\mathcal{S}^{d-1}$, then $\forall v \in \mathcal{S}^{d-1}, \exists j \in [K]$, s.t. $\left\| v - u^{(j)} \right\| \le \frac{1}{2}$. Hence we have

$$g_1^v(X) \le g_1^{u^{(j)}}(X) + |g_1^v(X) - g_1^{u^{(j)}}(X)| \le \max_j g_1^{u^j} + Z_1 \left\| v - u^{(j)} \right\|$$

As a result, $Z_1 \le 2 \max_{j=1,\cdots,K} g_1^{u^{(j)}}(X)$. Therefore it is sufficient to bound $g(X)$ for a fixed $u^{(j)} \in \mathcal{S}^{d-1}$. By Lemma A.1, covering number $K \le \exp(2d)$.

By Lemma 2, we have with probability at least $1 - \exp(-cd \log n)$, $g_1^{u_j} = \tilde{O}(\max\{R_n(\mathcal{F}_1^u), (1 + R_{\max})d/\sqrt{n}\})$. Plugging in the Rademacher complexity from Proposition 1, and applying union bound, we have

$$Z_1 \le 2 \max_j g_1^{u_j} \le \tilde{O}(\max\{n^{-1/2} M^3 (1 + R_{\max})^3 \sqrt{d} \max\{1, \log(\kappa)\}, (1 + R_{\max})d/\sqrt{n}\})$$

with probability at least $1 - \exp(2d - cd \log n) = 1 - \exp(-c'd \log n)$. $\qquad\square$

*Proof of Theorem 2.* We show the result by induction. When $t = 1$,

$$\left\| \boldsymbol{\mu}^1 - \boldsymbol{\mu}^* \right\|_2 = \left\| G_n(\boldsymbol{\mu}^0) - \boldsymbol{\mu}^* \right\| \le \left\| G(\boldsymbol{\mu}^0) - \boldsymbol{\mu}^* \right\| + \left\| G_n(\boldsymbol{\mu}^0) - G(\boldsymbol{\mu}^0) \right\|$$
$$\le \zeta \left\| \boldsymbol{\mu}^0 - \boldsymbol{\mu}^* \right\| + \epsilon^{\text{unif}}(n)$$

If $\left\| \boldsymbol{\mu}_i^t - \boldsymbol{\mu}_i^* \right\| < a$ and $\epsilon^{\text{unif}}(n) \le (1 - \zeta)a$, we have $\left\| \boldsymbol{\mu}_i^{t+1} - \boldsymbol{\mu}_i^* \right\| \le a$. So $\boldsymbol{\mu}^t$ lies in the contraction region for $\forall t \ge 0$.

Then iteratively we get

$$\left\| \boldsymbol{\mu}^t - \boldsymbol{\mu}^* \right\| \le \zeta \left\| \boldsymbol{\mu}^{t-1} - \boldsymbol{\mu}^* \right\| + \epsilon^{\text{unif}}(n)$$
$$\le \zeta^t \left\| \boldsymbol{\mu}^0 - \boldsymbol{\mu}^* \right\| + \sum_{i=0}^{t-1} \zeta^i \epsilon^{\text{unif}}(n)$$
$$\le \zeta^t \left\| \boldsymbol{\mu}^0 - \boldsymbol{\mu}^* \right\| + \frac{1}{1 - \zeta} \epsilon^{\text{unif}}(n)$$

with probability at least $1 - \delta$. $\qquad\square$

# D   Initialization

This section provides the number of initializations needed for the condition in Theorem 1.

**Proposition D.1.** *Let* $\pi_i = \frac{1}{M}, \forall i \in [M]$, $R_{\min} = \Omega(\sqrt{d})$, *and let a satisfy the conditions in Theorem 1. Then with* $\frac{\log(1/\delta)}{\sqrt{2\pi M}} \left( \frac{e}{1 - e^{-a\sqrt{d}/2}} \right)^M$ *initializations, the probability of having at least one good initialization is greater than* $1 - \delta$.

The proof follows directly from some combinatorial arguments and Lemma A.5.

*Proof of Proposition D.1.* Define event $\mathcal{E}_{init}(a) = \{\mu_i^0 \in \mathbb{B}_{\mu_i^*}(a), \; \forall i \in [M]\}$. By equal weights assumption, the probability of randomly sampled $M$ points having exactly one from each cluster is $\frac{M!}{M^M}$. By Sterling's formula, we have $M! \geq \sqrt{2\pi M}e^{-M}$. For each center, by Lemma A.5 we have the probability of it lying in $\mathbb{B}_{\mu_i^*}(a)$ is no less than $1 - e^{-a\sqrt{d}/2}$. Hence

$$P(\mathcal{E}_{init}(a)) \geq \sqrt{2\pi M}\left(\frac{1 - e^{-a\sqrt{d}/2}}{e}\right)^M =: p$$

Now assume the number of initializations is $T$, in order to satisfy the required property, we need $(1 - P(\mathcal{E}_{init}(a)))^T \leq \delta$. A sufficient condition is

$$T \geq \frac{\log(1/\delta)}{\log(1-p)}$$

Note that $\log(1 - x) \geq -x, \forall 0 \leq x \leq 0.5$. Since $p < .5$ for $M \geq 2$, we see that as long as $T \geq \frac{\log(1/\delta)}{\sqrt{2\pi M}}\left(\frac{e}{1-e^{-a\sqrt{d}/2}}\right)^M$, with probability $1 - \delta$ we will have a good initialization. $\qquad\square$

**Remark D.1.** *Perhaps not so surprisingly, the above theorem requires a stronger separation condition, i.e. $R_{\min} = \Omega(\sqrt{d})$, whereas all our analysis requires $R_{\min} = \Omega(\sqrt{d_0})$ where $d_0 := \min(d, M)$ can be thought of as effective dimension. This difficulty can be alleviated by using projections schemes similar to those in [1, 4]. We leave this for future work.*