[Reviews · NeurIPS 2017]

Reviewer 1



This paper studies the problem of learning mixtures of (spherical) Gaussians using EM algorithm. The main result is that under reasonable initialization, (population version of) EM algorithm converges efficiently to the optimal solution. The paper also gives corresponding finite sample results. The result relies on the gradient stability condition that is previously discussed in [1], but this result is more general compared to the Gaussian mixture result in [1] (which arguably is just an example instead of the main result). The proof involves a lot of calculations and bounding different terms. Overall this is an interesting paper. However the reviewer feels the paper completely ignored a long line of work that analyzes the k-means algorithm under very similar separating conditions. The line of work started at the paper by Kannan and Kumar "Clustering with Spectral Norm and the k-means Algorithm", and was improved by Awasthi and Or in "Improved Spectral-Norm Bounds for Clustering". In particular, the latter work analyzes k-means algorithm under the same (in fact slightly better) separation guarantee as in this paper. With this amount of separation the hard assignment (k-means) algorithm is not super different from the EM algorithm. Given these results this paper does not appear too surprising. Although the reviewer does note that the current paper is using a new proof technique. It's also worth mentioning that there are algorithms for mixture of spherical Gaussians that can work with even smaller separation (e.g. VempalaWang).

Reviewer 2



Summary: This paper derives statistical guarantees for the gradient EM algorithm applied to clustering in Gaussian mixture models, generalizing the results of [1] to multiple clusters and non-uniform cluster weights. The first result (Theorem 1) concerns the "population EM algorithm", i.e., when the expectation step can be computed exactly rather than being estimated from data. For this case, if the cluster centers are sufficiently separated and all the estimated cluster centers are initialized sufficiently close to the true cluster centers (i.e., roughly if there is already a sufficiently clear 1-1 correspondence between the estimated and true cluster centers), then the estimated cluster centers converge linearly to the true centers. This result relies on first showing (Theorem 4) that, under these conditions, due to properties of the Gaussian distribution, the curvature of the Q function is bounded (i.e., specifically, the gradient is Lipschitz). By a standard proof, the Lipschitz gradient then implies convergence of gradient descent. The paper then turns to showing that finite-sample gradient EM approximates population EM, based on a generalization [2] of McDiarmid's inequality (to unbounded data), and a novel bound on the Rademacher complexity of the gradient for GMMs. This in turn implies a linear convergence rate for finite-sample gradient EM, assuming sufficiently good initialization. Finally, the paper concludes with some synthetic experiments verifying the linear convergence rate and the necessity of the assumption that the initial clusters are good. Main Comments: This paper proves fairly strong, tight convergence bounds on a classic clustering algorithm, significantly generalizing the results of [1]. The proofs are well-motivated and lucidly sketched. I'm not too familiar with related work in this area, but I believe the results are quite novel. Overall, this paper is a significant contribution. However, the paper needs extensive proof-reading (see some of the typos noted below). Some thoughts: 1) The results rely on having a sufficiently good initialization of the estimated cluster centers. Although haven't thought in much detail, I suspect that (perhaps assuming the clusters centers are known to lie in a bounded region), it should be quite easy to derive a high-probability bound the number of random initializations needed to obtain such an initialization. Such a result might nicely complement the given results. 2) It is noted (Remark 4) that the altered version (from [2]) of McDiarmid's inequality used results in extra factors of sqrt(d) and log(n). I wonder if the recent variant (see [R1] below) of McDiarmid's inequality for sub-Gaussian data can give tighter results? This bound appears somewhat more similar to the original (bounded-differences) version of McDiarmid's inequality, though perhaps it is equivalent to that in [2]. The paper has a lot of typos and minor language issues: Line 18: "loglikelihood" should be "log-likelihood" Line 26 and Line 142: "convergence guarantees... is" should be "convergence guarantees... are" Line 27: "converge to arbitrarily" should be "converge to an arbitrarily" Line 30: The sentence "In [16]... is continuous." doesn't parse. Line 52: "semi definite" should be "semi-definite" Line 69: "Ledoux Talagrand" should be "Ledoux-Talagrand" Line 122: "defined in 1". What is "1"? Should this be the reference [1]? Lines 139-144: It looks like part of the related work section was randomly pasted into the middle of a sentence that should read "... contraction results [9] coupled with arguments that use..." Line 187: "equivalent as" should be "equivalent to" Hyphenation (e.g., "sub-Gaussian", "sub-optimal") is inconsistent throughout the paper. [R1] Kontorovich, Aryeh. "Concentration in unbounded metric spaces and algorithmic stability." Proceedings of the 31st International Conference on Machine Learning (ICML-14). 2014.

Reviewer 3



The paper studies convergence of gradient EM for isotropic Gaussian mixture models. They start with the traditional approach of first deriving population convergence and subsequently they analyze the finite sample case. However, they face a difficulty as the function class does not possess bounded differences. The authors circumvent this challenge by deploying an extension of McDiarmid's theorem. The authors obtain radius for contraction region and show some experiments on its tightness. The paper is well written and easy to understand, with a nice and gentle walk through to more complicated settings. Detailed proofs are provided in the supplementary materials, with some neat tricks. Unfortunately, adequate related works have not been reviewed. Especially, since the paper analyses the convergence properties for isotropic Gaussian Mixture models, the need to compare to the k-Means problem is inevitable. Comments: 1. I could not find the definition of G (caps G) in the paper. I suspect it represents the gradient update operator as defined in equation (2). Further, "s" is not defined or is mentioned that s is positive in equation (2), which I believe is the step-size as referenced much later. Please clarify and update in the paper. 2. Line 144 has a few words missing: "sub-optimal point. uments that use" 3. In the case of the isotropic Gaussian mixture model, full EM update is no costlier than the full gradient update. What is the motivation of applying gradient EM? A stochastic gradient EM would be highly valuable though, like analyzed in [1]. Does the current analysis carry over to stochastic gradient case? 4. Finally, since the paper focuses wholly on isotropic Gaussian mixture models, it warrants a comparison to the k-Means problem in L2 distance. For example, we know that PTAS is available for k-Means for arbitrary initialization (cf Z. Friggstad, M. Rezapour, and M. R. Salavatipour Local Search Yields a PTAS for k-Means in Doubling Metrics FOCS, 2016. and Jiri Matousek. On approximate geometric k-clustering. Discrete & Computational Geometry, 24(1):61–84, 2000.). However, as shown in the paper the contraction region for the gradient EM is less than R_min/2. This region can be very small and it is hard to find an initialization here. Can local search help in expanding the contraction region? Overall the paper presents a neat convergence analysis, an interesting derivation, and a near optimal contraction region, albeit with no motivation. As mentioned above providing some motivation or studying the case of stochastic gradients should make the paper strong and a nice contribution to the field.